# Unravelling Heavy Metal Dynamics in Soil and Honey: A Case Study from Maramureș Region, Romania

**DOI:** 10.3390/foods12193577

**Published:** 2023-09-26

**Authors:** Florin Dumitru Bora, Anca Cristina Babeș, Anamaria Călugăr, Mugurel Ioan Jitea, Adela Hoble, Răzvan Vasile Filimon, Andrea Bunea, Alexandru Nicolescu, Claudiu Ioan Bunea

**Affiliations:** 1Viticulture and Oenology Department, Advanced Horticultural Research Institute of Transylvania, Faculty of Horticulture and Business in Rural Development, University of Agricultural Sciences and Veterinary Medicine Cluj-Napoca, 3-5 Mănăștur Street, 400372 Cluj-Napoca, Romania; boraflorindumitru@gmail.com (F.D.B.); babesanca@yahoo.com (A.C.B.); anamaria.calugar@usamvcluj.ro (A.C.); 2Laboratory of Chromatography, Advanced Horticultural Research Institute of Transylvania, Faculty of Horticulture and Business for Rural Development, University of Agricultural Sciences and Veterinary Medicine, 400372 Cluj-Napoca, Romania; alexandru.nicolescu@usamvcluj.ro; 3Department of Economic Sciences, Faculty of Horticulture and Business in Rural Development, University of Agricultural Sciences and Veterinary Medicine Cluj-Napoca, 3-5 Mănăștur Street, 400372 Cluj-Napoca, Romania; mjitea@usamvcluj.ro; 4Research Laboratory Regarding Exploitation of Land Improvement, Land Reclamation Systems and Irrigation of Horticultural Crops, Advanced Horticultural Research Institute of Transylvania, Faculty of Horticulture and Business in Rural Development, University of Agricultural Sciences and Veterinary Medicine Cluj-Napoca, 3-5 Mănăștur Street, 400372 Cluj-Napoca, Romania; adela.hoble@usamvcluj.ro; 5Research Development Station for Viticulture and Winemaking Iași, 48 Mihail Sadoveanu Alley, 700490 Iasi, Romania; razvan_f80@yahoo.com; 6Biochemistry Department, Faculty of Animal Science and Biotechnology, University of Agricultural Sciences and Veterinary Medicine Cluj-Napoca, 3-5 Mănăștur Street, 400372 Cluj-Napoca, Romania; andrea.bunea@usamvcluj.ro; 7Department of Pharmaceutical Botany, Faculty of Pharmacy “Iuliu Hațieganu”, University of Medicine and Pharmacy, 23 Gheorghe Marinescu, 400337 Cluj-Napoca, Romania

**Keywords:** heavy metal contamination in honey, food quality analysis, polluted areas, honeybee population, environmental biomonitoring

## Abstract

The study examined soil and honey samples from the Maramureș region, assessing potentially toxic elements and their concentrations. The highest concentrations were found for (Cu), (Zn), (Pb), (Cr), (Ni), (Cd), (Co), and (As), while (Hg) remained below the detection limit. Samples near anthropogenic sources displayed elevated metal levels, with the Aurul settling pond and Herja mine being major contamination sources. Copper concentrations exceeded the legal limits in areas near these sources. Zinc concentrations were highest near mining areas, and Pb and Cd levels surpassed the legal limits near beehives producing acacia honey. Nickel and Co levels were generally within limits but elevated near the Herja mine. The study highlighted the role of anthropogenic activities in heavy metal pollution. In the second part, honey samples were analyzed for heavy metal concentrations, with variations across types and locations. Positive correlations were identified between certain elements in honey, influenced by factors like location and pollution sources. The research emphasized the need for pollution control measures to ensure honey safety. The bioaccumulation factor analysis indicated a sequential metal transfer from soil to honey. The study’s comprehensive approach sheds light on toxic element contamination in honey, addressing pollution sources and pathways.

## 1. Introduction

Over the past decade, driven by heightened public concern for food safety and recognizing the vital role of food in promoting a healthy human diet, ensuring food safety has emerged as a pivotal aspect of food quality [1,2]. Among the staples of human diets—such as meat, vegetables, fish, and grains—lie crucial sources of vitamins, minerals, proteins, and carbohydrates that nourish the body [3]. Consequently, the consumption of diverse foods contributes significantly to nutrient intake, while concurrently serving as a potential pathway for pollutants to enter the human body [4,5,6,7].

As per the international Codex Alimentarius Commission, honey is a natural sweet substance synthesized by honeybees (*Apis mellifera* L.) through the collection of nectar from flowers or secretions of living plants [8,9,10].

Key quality indicators for honey encompass its physicochemical attributes, mineral composition, and heavy metal concentrations. If the values of these indicators exceed national and international recommendations, potential health issues can arise, such as disruptions in iron transport, cellular adhesion, mutagenic and carcinogenic effects, and instances of toxicity and oxidative stress [11]. The quantity and quality of minerals are distinct for each plant’s flower in each region of the country; hence, the overall mineral content is influenced by the geographical location [12]. Additionally, the quality of the honey is directly influenced by factors like the harvesting time, storage conditions, and storage locations [10].

To ensure the authenticity of honey and to enhance its production capacity to meet global market demand, it is imperative to comprehensively analyze the physicochemical properties, mineral content, and heavy metal presence in honey [13].

The composition and characteristics of honey are directly shaped by both biotic and abiotic factors. These factors encompass aspects such as the specific type of flora, the techniques employed in honey processing and storage, and the prevailing environmental conditions in which the plants have thrived [14]. Bees, due to their direct interaction with the air, soil, and water, particularly during their foraging endeavors, cover extensive foraging territories, which can extend beyond 7 km^2^ [15] and, in some cases, even reach up to 100 km^2^ [16]. This exposes them to a diverse array of potential contaminants, including potentially toxic elements (PTEs) or heavy metals (HMs), which they may carry back to their hives [15].

Taking these factors into account, honey holds the potential to serve as an indicator of environmental contamination, reflecting the presence of substances like pesticides, potentially toxic elements (PTEs), and radioactivity [17]. The occurrence of metals within honey has been linked to the placement of beehives in proximity to pollution sources, such as tailings ponds, mines, highways, factories, or regions characterized by significant volcanic activity. In addition to these pollution sources, the contamination can also stem from agrochemicals that contain substances like cadmium and arsenic, among others [18,19,20,21,22,23].

The European honey bee, Apis mellifera, stands out as a valuable tool in pollutant biomonitoring efforts involving insects. As a primary pollinator of agricultural systems, this bee species engages with the environment extensively and spans a global distribution [21]. While individual bees may be negatively affected by environmental stressors due to their foraging activities, bee colonies tend to exhibit greater resilience and can accumulate or respond to stressors without collapsing [21]. With tens of thousands of foraging bees in a colony, they serve as sampling agents of the surrounding environment. Bees can come into contact with contaminants at pollination sites or during flights to and from the hive. Research by Zarić et al. (2017) [24] highlighted how foraging bees collect heavy metals from vegetation, contaminated water, and airborne particulates, which adhere to their body hairs. Upon returning to their colonies, these metals can be found in stored pollen, beeswax, honey, and propolis, a resin-like substance harvested from trees [21]. While heavy metals can impact brood production, survival rates, or navigation skills in bees [25], their accumulation in honey bees and their hives is not lethal to the colony and presents an avenue for environmental monitoring [21].

Examining the existing scientific literature on metal concentrations in the environment and bees reveals a focus on metals with potentially toxic effects. Numerous studies have explored various aspects of the interplay between the toxic metal content in the environment, bees, and bee products [26,27,28,29,30]. Many of these studies propose the utilization of bees and their products as bioindicators of toxic metal pollution [29,31] and highlight their potential in biomonitoring processes [27,32,33,34,35,36,37,38]. Some research delves into the impact of pollen on the metal profile in consumed honey [39], while other studies evaluate the human health risks stemming from the intake of toxic metals present in honey [40,41]. In recent years, more attention has been directed toward understanding the influence of toxic metals on bee development and survival, as well as the physiological and biochemical changes underlying the adverse effects of certain heavy metals [41]. However, only a relatively small number of researchers have addressed the concentrations of essential bioelements and their impacts on bee physiology [42].

Maramureș County, situated in northern Romania, boasts a history of non-ferrous mineral extraction that has played a significant role in the region’s economy for centuries [43]. Following the Great Union in 1918, the primary focus of metal mining in the Baia Mare area was on the extraction and processing of gold, silver, lead, copper, zinc, and pyrites [43]. The extracted materials underwent initial ore concentration processes in designated spaces known as purification stations—a widely used operation in the mining industry worldwide [43]. The flotation methods used led to the accumulation of waste, contributing directly to the formation of tailings ponds [43]. However, the mining operations and waste deposits ceased activity due to Romania’s non-compliance with environmental commitments stipulated by the Treaty of Accession to the European Union on 1 January 2007 [43]. On a broader scale, soil contamination by heavy metals has become a significant global concern, with notable issues of soil pollution also present in Romania. In fact, the country has identified 108 tailings ponds and 1101 sterile dumps, with 180 sterile dumps located in Maramureș County [44].

This research delves into the use of honeybees as bioindicators to assess the levels of specific elements in soil and honey, with a focus on potentially toxic elements, at sampling sites characterized by diverse environmental impacts within the Maramureș–Baia Mare region of Romania. The underlying premise is that the Baia Mare region is recognized not just for its underground mineral wealth but also for being one of Romania’s most polluted areas due to intensive mining activities involving the extraction of precious metals and ore processing containing Cd, Cu, Pb, and Zn [45]. Furthermore, this study aims to compare Romania’s overall environmental health status in terms of heavy metal contamination by employing living organisms.

## 2. Materials and Methods

### 2.1. Research Location

A total of thirty-eight soil samples, designated as S1–S11, and thirty-eight *Apis mellifera* honey samples, denoted as H1–H11, were gathered in 2017 and 2021 from nine distinct geographical regions within the northern part of Romania. These provinces include Tăuții de Sus, Tăuții Măgherăuș, Baia Mare, Baia Sprie, Baia Borșa, Satulung, Săcălășeni, Groșii Țibleșului, and Vișeu de Sus. The honey samples consisted of acacia honey (*n* = 3), chestnut honey (*n* = 13), and multifloral honey (*n* = 22). For additional details regarding the study area and primary sources of heavy metal pollution (anthropogenic sources of heavy metal pollution), as well as the relatively less contaminated zone within the Maramureș county (background area), please refer to Figure 1. The beehives were strategically situated away from pollution sources, making this specific region the selected control or background area.

### 2.2. Collection of Soil and Honey Samples

#### 2.2.1. Surface Soil Sampling

Soil samples from nine locations were collected using an opportunistic sampling method, as described by Pennock et al. (2007) [46], during the spring–summer seasons of 2017 to 2020. Each year, approximately 0.5 kg of soil was collected from non-agricultural land at each sampling point. This involved gathering four to seven sub-samples from an area of around 100 × 100 m. The sampling sites were chosen based on accessibility and with proper authorization from relevant authorities or property owners. All topsoil samples were retrieved from designated research hive sites, including public spaces and residential gardens. The uppermost 10 cm of soil, just beneath the recent layer of vegetation (if present), was collected using a polyvinyl chloride (PVC) corner and plastic trowel (ISO 11464/1994). Care was taken to exclude large aggregates and debris. Once collected, the soil samples were transported to the laboratory. Roots and stones were eliminated, and the soil was thoroughly mixed to ensure sample homogeneity. Appendix A provides additional details about the geographic origin of the soil samples, sampling depth, anthropogenic impact, and the approximate distance of the hives from pollution sources.

#### 2.2.2. Honey Sampling

The honey samples were directly obtained from beekeepers during the summer seasons of 2017–2021 (June–August). The honey extraction process involved centrifugation of the combs. All collected samples were in their unpasteurized form and were gathered within a maximum period of three months post-extraction. These samples were stored in hermetically sealed glass containers until the chemical analyses were conducted, and they were kept in a cool and dark environment at temperatures ranging between 4 and 5 °C. The honey used for analysis was exclusively sourced from the *Apis mellifera* species. Appendix A provides additional details about the honey samples, including their geographical origin, harvest time frame, the extraction method employed, and the estimated distance of the hives from potential pollution sources.

### 2.3. Sample Preparation and Microwave Digestion Procedure

#### 2.3.1. Preparation Soil Sample

The topsoil samples were subjected to a series of preparatory steps. Initially, they were subjected to oven-drying at 105 °C for a minimum of 72 h, using a Binder FD 53 oven (Darmstadt, Germany). Subsequently, the dried samples were sieved and disaggregated through a nylon 2 mm screen using an automated Retsch 110 mill (Darmstadt, Germany). The resulting dried and sieved topsoil was then homogenized and subsampled. To further ensure uniformity, the samples were manually powdered in an agate mortar and pestle. The soil sample preparation methodology had been established in a previous study [47].

For the microwave digestion process, which was optimized in earlier research [47], approximately 0.2–0.5 g of the dried and milled soil sample was directly weighed into a clean Teflon digestion vessel. Subsequently, 12 mL of aqua regia (9 mL HCl + 3 mL HNO_3_) and 1 mL of H_2_O_2_ were added to the vessel. After allowing a 15 min period for pre-digestion, the actual mineralization process was carried out using the Milestone START D Microwave Digestion System. The specific digestion program utilized is detailed in Appendix A. Once the mineralization was completed, the Teflon digestion vessel was opened under the strictest safety protocols. The resultant solution was then filtered using a 0.45 µm PTFE membrane filter. The filtrate was quantitatively transferred to a 50 mL volumetric flask and supplemented with deionized water to reach the mark after cooling to room temperature.

#### 2.3.2. Preliminary Preparation of Honey Sample

To ensure homogeneity, sealed honey samples (preferably in their original containers) were placed in a rotating water bath and heated at 60 °C for 30 min. Throughout the heating process, the samples were shaken and mixed to enhance uniformity. Because honey possesses solubility in water and metals can be directly quantified in solutions produced by dissolving honey samples in water or acidic solutions, it is customary to break down honey samples before analysis [48]. The resulting residues, whether stemming from ash or digestion, are subsequently re-dissolved in an acid solution (or ultrapure water). This process facilitates the transfer of mineral constituents into the solution, disrupting the organic matrix of honey and enabling the extraction of metal species [49].

For chestnut, acacia, and polyfloral honey samples devoid of visible granules, homogenization was achieved through shaking. Honey samples containing crystallized sugar (seven in total) were subjected to heating at 60 °C in a water bath for 30 min to dissolve the crystals and promote uniformity. Approximately 1 g of homogenized honey was measured and placed into polypropylene tubes. These samples were dissolved in 20 mL of deionized ultra-pure water heated to 60 °C, characterized by a specific resistivity of 18.2 MΩ × cm^−1^, utilizing the Milli-Q Integral Ultrapure Water-Type 1.

Subsequently, the solutions obtained underwent mineralization through a Milestone START D Microwave Digestion System (Sorisole, Italy). In this process, about 0.5 g of honey sample was directly weighed into a clean Teflon digestion vessel. Subsequently, 12 mL of aqua regia (comprising 9 mL HCl and 3 mL HNO_3_) was added. Following a 15 min interval, mineralization was executed using the Milestone START D System.

### 2.4. Basic ICP-MS Analytical Instrumental Parameters

Analytical measurements of microelements (^63^Cu, ^64^Zn), ultra-trace elements (^52^Cr, ^59^Co, ^60^Ni), and heavy metals (^75^As, ^111^Cd, ^201^Hg, ^208^Pb) were conducted using an ICP-MS (iCAP Q ICP-MS Thermo Fisher Scientific, Waltham, MA, USA). The instrument was equipped with an ASX-520 autosampler, a micro-concentric nebulizer, a Ni sampler and Ni skimmer cones, and a peristaltic sampled delivery pump, operating in quantitative analysis mode. Sample solutions were introduced into the ICP-MS plasma using a nebulizer attached to a cyclonic spray chamber, while the standard ICP-MS torch included a 1.5 mm diameter injector. The analysis method employed in this study was inductively coupled plasma mass spectrometry (ICP-MS), incorporating collision cell technology for the efficient elimination of common ICP-MS interferences using pure helium as the collision cell gas and kinetic energy discrimination.

Before the quantitative assessment of samples, the ICP-MS system was allowed to equilibrate for a minimum of 45 min, during which the experimental conditions and mass calibration were verified. A short-time stability test was performed using a tuning standard solution (TUNE B iCAP Q Ba, Bi, Ce, Co, In, Li, U, 1.0 μg/L (each) in 2% HNO_3_ + 0.5 HCl) encompassing the entire mass range. This auto-tuning process optimized the plasma sampling zone for a balance between high sensitivity, ion optics voltage optimization, and minimal levels of cluster ions and doubly charged ions. The ICP-MS was daily optimized to maximize sensitivity for M+ ions, with the monitoring of double ionization and oxides through the Ba^2+^/Ba^+^ and Ce^2+^/CeO^+^ ratios, respectively, both of which remained below 2%. The argon (Ar 5.0) 99.999 and helium (He 6.0) 99.9999% gases used purity (Messer, Austria). Each sample underwent duplicate analysis, with each analysis consisting of seven replicates. The detailed operational parameters of the ICP-MS can be found in Appendix A. The general ICP-MS instrumental parameters analysis has been previously presented and optimized in prior work [50].

### 2.5. Chemicals and Apparatus

All chemicals and reagents utilized in the experiments were sourced from reputable suppliers (Merk and Sigma Aldrich, Darmstadt, Germany) and possessed high purity. A solution of 65% HNO_3_ of supra-pure quality for trace analysis (Merk, Darmstadt, Germany) and a 30% H_2_O_2_ solution for trace analysis (Sigma Aldrich, Darmstadt, Germany) were employed. External standard calibration was employed for analysis after proper dilution. For this purpose, Ge, Tb, Rh, and Sc in supra-pure 1% HNO_3_ (Merk, Darmstadt, Germany) were utilized as internal standards. The internal standard, added at a level of 50 µg/L, was incorporated into all samples, including blanks and standards.

The calibration curve was generated and calibrated using high-purity ICP Multi-Element Standard Solution XXI CertiPUR (Merk, Darmstadt, Germany). The calibration method, along with the utilization of the internal standard, was developed in previous research [51]. Aqueous solutions were prepared by blending standards with high-purity deionization water. The (pre)preparation of samples and solutions involved ultrapure water obtained from Milli-Q Integral Ultrapure Water-Type 1 (with a specific resistivity of 18.2 MΩ × cm^−1^).

Teflon digestion vessels underwent cleaning with 25 mL HNO_3_ before each mineralization process, and the entire mineralization process for soil and honey samples was conducted in triplicate. The digestion system accommodated a maximum of six digestion vessels (five vessels for samples and one for blank), constructed from modified polytetrafluoroethylene (TFM-PTEE). All flasks employed in the experiments were rinsed with 5 M HNO_3_ for 24 h and then thoroughly washed with deionized water three to four times. The high-precision analytical balance KERN ADB 100-4 was utilized for weighing soil and honey samples, as well as for the preparation of working and calibration solutions.

### 2.6. Chemical Analysis Quality Control

The determination of the limits of detection (LoDs) and the limits of quantification (LoQs) was conducted in accordance with Commission Regulation (EU) No. 2016/582 of 15 April 2016, which amended Regulation (EC) No. 333/2007. This regulation outlines the analysis of inorganic arsenic, lead, polycyclic aromatic hydrocarbons, and specific performance criteria for analysis [51]. The LoDs and LoQs for the analyzed elements were calculated based on the standard deviation (σ) derived from 20 independent measurements of a blank solution, with 3σ for the LoDs and 10σ for the LoQs (Appendix A).

The repeatability was assessed using the Horwitz ratio (HorRat), which involved dividing the values of the relative standard deviation (RSDr) by the RSDs values estimated from the Horwitz equation [52]. All resulting values needed to be below 2, ensuring the reliability of the method. The validation parameters of the analytical procedures, including the precision, accuracy, recovery, and uncertainty, are presented in Appendix A.

The calibration standards were prepared from the ICP Multi-Element Standard Solution XXI CertiPUR in five different concentrations (2.5, 5, 10, 25, and 50 µL). To assess the accuracy and precision, the analytical procedure involved spiking a known amount of the analyte metal into a test portion of the sample. This test portion was then analyzed alongside the original sample. The method’s precision was expressed as the percent relative standard deviation (RSD%) from triplicate analyses.

Furthermore, recovery assays for soil and honey samples at a concentration of 5 µL were conducted. Three replicates of this concentration level (*n* = 3) resulted in average recovery values ranging from 85.3% to 117.89%.

### 2.7. Statistical Analysis

Data collection and descriptive statistics, including the calculations of averages, medians, relative standard deviations, Spearman’s correlation, and bioaccumulation factors, were conducted using Microsoft Excel 365 (Microsoft, New York, NY, USA) and Addinsoft version 15.5.03.3707 (Microsoft, New York, NY, USA). The data precision was assessed and presented as the standard deviation (SD). All data were reported as means ± standard deviations and underwent statistical analysis using SPSS Version 24 (SPSS Inc., Chicago, IL, USA). The statistical outcomes were presented as means (averaged over three replications) and standard deviations. The recorded data were subjected to a two-way analysis of variance (ANOVA) to explore the impact of various variables on the concentrations of heavy metals in both soil and honey samples. Further analysis of variance (ANOVA) and average separation were performed using the Duncan test at a significance level of *p* ≤ 0.005 with SPSS Version 24 (SPSS Inc., Chicago, IL, USA).

## 3. Results

### 3.1. Soil

In the Maramureș region, a total of thirty-eight soil samples were assessed for nine different elements at depths of 0–10 cm (refer to Table 1). The analysis of the heavy metal concentrations in the soil samples around the beehives revealed significantly elevated values compared with the maximum permissible limits set by both the Romanian Regulation for allowable levels of hazardous and harmful substances in soil (Order of the Ministry of Waters, Forest and Environmental Protection No. 765/3 November 1997) and the Council Directive 86/278/EEC for the Protection of the Environment (European Communities Council 1986) [53].

The concentration of mercury (Hg) in all analyzed samples was found to be below the detection limit quantification (LoQ for Hg: 0.1379 µg/L) of the employed analytical method. Further details regarding the impact of environmental factors on the accumulation of heavy metals in soil and honey can be found in Appendix A.

The concentration of copper (Cu) in soil samples collected near the heavily trafficked European Road (E 58) showed normal values in both the Satulung (3.39 mg/kg) and Săcălășeni (0.78 mg/kg) study areas. In this context, there was no indication of a negative impact from car traffic on the Cu concentration in the soil. When examining the distribution of the Cu concentration in areas unaffected by pollution versus those influenced by anthropogenic factors, the element exhibited a value of 978.98 mg/kg in areas influenced by such factors, contrasting with a value of 2.05 mg/kg in unpolluted regions. Notably, in the background zone, the concentration of Cu was relatively low compared with the other studied areas. The obtained Cu results aligned with findings from previous studies. For instance, Huzum et al. (2012) [54] reported similar values (256.00 mg/kg) from research conducted on soil cultivated with vines in the Huși area, Romania. Similarly, Bora et al. (2020) [50] documented ranges of 621.79–4155.94 mg/kg in soil cultivated with vines from the Baia Mare area, Romania. In the Baia Mare area, Chakraborty et al. (2017) [55] found concentrations spanning 19.8–2760.0 mg/kg in soil polluted with heavy metals. Paulette et al. (2015) [56] identified concentrations ranging from 77 to 7675 mg/kg in soil polluted with heavy metals in the Copșa Mică area, Romania. Mihali et al. (2017) [57] observed levels from 40.9 to 621.6 mg/kg in soil polluted with heavy metals from the Baia Mare area. Albulescu et al. (2009) [58] reported values between 36.63 and 112.00 mg/kg in soil cultivated with vines from the Caraş-Severin area. Finally, Bora et al. (2015) [47] recorded a concentration of 479.64 mg/kg in soil polluted with heavy metals near the Baia Mare area, Romania.

The distribution of the zinc (Zn) concentration within the studied area revealed that the highest levels were identified in Tăuții Măgherăuș (ranging from 2834.58 to 2588.59 mg/kg) for soil samples collected between 2018 and 2019, as well as in Baia Mare near hives producing acacia honey (ranging from 2533.76 to 2379.58 mg/kg) (see Table 1). Elevated Zn concentrations were also observed in other investigated regions, which are known to have sources of heavy metal pollution. For instance, in Tăuții de Sus near hives producing chestnut honey, the average Zn concentration stood at 1479.51 mg/kg. Similarly, Tăuții Măgherăuș, near hives producing chestnut honey, exhibited an average Zn concentration of 2438.40 mg/kg. Baia Mare, near hives producing chestnut honey, had an average Zn concentration of 1731.69 mg/kg, while Baia Sprie, near hives producing polyfloral honey, had an average concentration of 1317.23 mg/kg (refer to Figure 1). These areas are impacted by the presence of the Aurul settling pond in Tăuții de Sus, the Bozânta Mare settling pond, and the Herja mine. The measured values surpassed the permissible maximum limits of 100 mg/kg for Zn. In this study, the highest recorded value was in Tăuții Măgherăuș at 2834.58 mg/kg Zn, with an average value of 2438.40 mg/kg Zn, regardless of the area or year of sample collection. These elevated readings could be attributed to the Baița mine, situated approximately 11.0 km from the collection site in Tăuții Măgherăuș, and the Aurul decantation pond in Tăuții de Sus, positioned around 8.0 km from the sample collection site in Baia Mare. The outcomes for the Zn concentration were in line with those found in prior studies. For example, Huzum et al. (2012) [54] reported similar levels at 61.10 mg/kg. Bora et al. (2020) [50] found a range of 45.36 to 3483.25 mg/kg, while Chakraborty et al. (2017) [55] noted concentrations from 54.4 to 2370.0 mg/kg. Mihali et al. (2017) [57] documented levels between 82.26 and 1002 mg/kg. It is worth noting that the results presented by Bora et al. (2015) [47] at 69.44 mg/kg were considerably lower than those observed in this current research.

For lead (Pb) and cadmium (Cd) concentrations, these heavy metals exhibited elevated levels in the Baia Mare and Baia Sprie regions, particularly in soil samples taken from the vicinity of beehives producing acacia honey. Specifically, in the years 2020 and 2021, Pb concentrations were measured at 1205.57 ± 70.56 mg/kg and 1166.11 ± 68.21 mg/kg, respectively, resulting in an average of 1185.84 mg/kg. Meanwhile, Cd concentrations during the same years were found to be 6.33 ± 1.57 mg/kg and 5.52 ± 1.90 mg/kg, with an average of 5.93 mg/kg (refer to Table 1). Further analysis of the results indicated that soil samples collected near beehives producing chestnut honey in Baia Mare and Baia Sprie also exhibited elevated levels of Pb and Cd compared with other study areas. These measured values surpassed the maximum limits established by regulations for both Pb and Cd. One possible explanation for these excesses could be the adverse impact of the Aurul settling pond in Tăuții de Sus on the environment. Conversely, in regions with reduced anthropogenic influence such as areas near the heavily trafficked European Road (E 58) or the former mining flotation site in Băile Borșa, the levels of Pb and Cd in soil samples were lower, indicating a lesser impact of these human activities on the environment. As expected, the levels of Cd and Pb in soil collected from the background area were below the detection limit. An exception to this was in Groșii Țibleșului for Pb, where values above this limit were recorded (0.02 mg/kg in soil samples from 2020), as well as in the Vișeu de Sus area (0.02 mg/kg in soil samples from 2019).

The results obtained for lead (Pb) and cadmium (Cd) were in line with findings from previous studies, including Huzum et al. (2012) [54] (12.90 mg/kg Pb, 0.21 mg/kg Cd), Bora et al. (2020) [50] (ranging from 6.62 to 4262.23 mg/kg Pb, 0.12 to 32.53 mg/kg Cd), Chakraborty et al. (2017) [55] (ranging from 38.0 to 14,329.0 mg/kg Pb), Paulette et al. (2015) [56] (ranging from 705 to 10,074 mg/kg Pb), Mihali et al. (2017) [57] (ranging from 48.12 to 3472 mg/kg Pb, 0.04 to 11 mg/kg Cd), Albulescu et al. (2009) [58] (21.90 mg/kg Pb, 1.77 mg/kg Cd), and Bora et al. 2015 [47] (14.77 mg/kg Pb, 0.45 mg/kg Cd). The mean values for nickel (Ni) and cobalt (Co) in soils sampled from the studied areas were 2.58 mg/kg Ni and 0.85 mg/kg Co, with a range from 0.42 mg/kg in 2021 to 7.99 mg/kg in 2019 for Ni and from 0.25 mg/kg in 2020 to 5.98 mg/kg in 2020 for Co (see Table 1). The highest Ni values were observed in samples collected near the Herja mine (~8.0 km from the Herja mine). Similarly, for Co, the maximum values were recorded in soil samples collected near the Aurul settling ponds in Tăuții de Sus (~8.0 km from the Aurul settling ponds in Tăuții de Sus). When comparing the maximum Ni and Co values with national legislation, it was evident that both elements remained below the legal limits.

In regions near the heavily trafficked European Road (E 58), such as Satulung and Săcălășeni, the concentrations of Ni and Co in soil samples were low. From the provided data, it can be concluded that the influence of car traffic on the concentration of these elements in these areas was minimized. The Aurul settling pond in Tăuții de Sus significantly affected the Ni and Co concentrations in soil samples collected from the same area, yielding an average of 1.97 mg/kg Ni and 0.31 mg/kg Co. Similar influences were observed in Tăuții Măgherăuș due to the former Nistru and Băița mines, as well as in the Baia Borșa area due to the former mining operations there. The results for Ni and Co closely aligned with findings from prior research. For instance, Huzum et al. (2012) [54] reported values of 29.90 mg/kg Ni and 7.20 mg/kg Co, while Bora et al. (2020) [50] found ranges of 6.97 to 28.60 mg/kg Ni and 5.08 to 29.57 mg/kg Co. Mihali et al. (2017) [57] observed ranges of 4.98 to 9.09 mg/kg Ni and 3.3 to 8.2 mg/kg Co. Albulescu et al. (2009) [58] reported 24.55 mg/kg Ni, and Bora et al. 2015 [47] documented 16.28 mg/kg Ni and 1.47 mg/kg Co.

Regarding arsenic (As) and chromium (Cr) concentrations, these heavy metals exhibited elevated levels in the Baia Mare area, specifically in soil samples collected from the vicinity of beehives producing chestnut honey. The measurements for As were 4.37 ± 1.85 mg/kg in 2020, 4.09 ± 0.50 mg/kg in 2019, and 3.60 ± 0.10 mg/kg in 2021, resulting in an average value of 0.89 mg/kg. Similarly, Cr concentrations for 2021 were 10.75 ± 3.01 mg/kg, for 2020 they were 10.71 ± 2.83 mg/kg, and for 2019 they were 10.07 ± 1.89 mg/kg, leading to an average value of 3.66 mg/kg (as indicated in Table 1). Substantial As and Cr concentrations were also found in soil samples taken near the beehives where acacia honey was produced in the Baia Mare area (2.35 mg/kg As average value) and near the Tăuții de Sus area (7.12 mg/kg Cr average value), where chestnut honey was obtained. The results highlight that soil samples collected near beehives producing chestnut and acacia honey in the Tăuții de Sus and Baia Mare areas exhibited notably higher As and Cr values compared with other study areas. The elevated concentrations of As and Cr in these areas can be attributed to the presence of the Aurul settling pond in Tăuții de Sus and the nearby settling pond in the Baia Mare area.

When comparing the maximum values achieved for arsenic (As) and chromium (Cr) with the regulations set by national legislation, it was evident that both elements remained within the limits stipulated by the law. The levels of As and Cr in soil samples from Satulung and Săcălășeni, which were situated near the heavily trafficked European Road (E 58), were relatively low. Based on the provided data, it can be inferred that the impact of car traffic on the concentration of these elements was minimized in these areas. The results obtained for As and Cr closely paralleled findings from previous studies. For instance, they aligned with values reported by Huzum et al. (2012) [54] (11.20 mg/kg As, 208.00 mg/kg Cr), Bora et al. (2020) [50] (ranging from 1.15 to 5.13 mg/kg As, 2.75 ± 0.65 mg/kg Cr), Mihali et al. (2017) [57] (ranging from 0.61 to 80.1 mg/kg As), and Albulescu et al. (2009) [58] (13.32 mg/kg Ni).

### 3.2. Honey

Table 2 presents the mean concentrations along with their corresponding standard deviations for various heavy metals found in three distinct types of honey: chestnut, acacia, and polyfloral. These honey samples were gathered from various locations over several years, all of which had been exposed to different levels of contamination from various human-made sources of heavy metals. To contextualize these findings, a comparison was made with the maximum permissible contaminant levels for heavy metals in both food (as specified by Commission Regulation (EC) No 1881/2006) and honey (according to Codex Alimentarius and Council Directive 2001/110/EC). The honey samples were collected from the same locations as the soil samples. Following the analysis, the heavy metals were ranked by their mean concentrations (in mg/kg) as follows: chromium (Cr) with the highest mean concentration (0.58 mg/kg), followed by zinc (Zn) (0.56 mg/kg), copper (Cu) (0.34 mg/kg), lead (Pb) (0.10 mg/kg), nickel (Ni) (0.03 mg/kg), cadmium (Cd) (0.01 mg/kg), and cobalt (Co) (below the limit of detection, with a limit of quantification of 0.136 µg/L). Arsenic (As) and mercury (Hg) were also below their respective limits of detection, with limits of quantification of 0.743 µg/L and 0.138 µg/L, respectively.

Samples were collected from a total of thirty-nine distinct locations for the analysis of copper (Cu) concentration in honey. The observed Cu levels ranged from 0.10 mg/kg (averaging across polyfloral honey samples from Groșii Țibleșului) to 0.79 mg/kg (noted in polyfloral honey samples from Baia Sprie). In other locations, the Cu values fell below the detection limit (LoQ for Cu: 1.545 µg/L). Notably elevated Cu concentrations were also detected in honey samples from the Baia Mare area. Here the recorded values ranged from 0.47 mg/kg (in polyfloral honey from 2021) to 0.56 mg/kg (observed in the same type of honey from 2020), with an average concentration of 0.52 mg/kg.

Based on the findings, the heightened Cu values in honey samples from the Baia Sprie and Baia Mare regions were likely attributable to the historical Herja mine and the Aurul settling pond in Tăuții de Sus. The proximity of these sources to the hive locations contributed to these elevated concentrations. It is noteworthy that the Cu levels detected in the Baia Mare and Baia Sprie samples surpassed the maximum permissible limit established by international regulations, which was set at 0.50 mg/kg. In light of this, it is advisable to consider relocating beehives away from areas closely associated with major sources of heavy metal pollution, such as the former Herja mine and the Aurul settling pond in Tăuții de Sus.

While copper (Cu) is essential for the health of various living organisms, including humans, it is important to acknowledge that an excess of Cu has been linked to adverse effects like cellular and tissue damage (as seen in Wilson’s disease) and various human disorders. Therefore, it is imperative to consider daily Cu intake from different sources, including food [60].

Additionally, it is noteworthy that acacia honey exhibited the highest Cu concentration (with an average value of 0.52 mg/kg) compared with the other types of honey under study, including chestnut honey (average: 0.35 mg/kg) and polyfloral honey (average: 0.18 mg/kg). This elevated Cu content can be attributed to the relatively close proximity of the beehives to former mining sites such as Herja, Nistru, Băița, and the mining flotation operations in Băile Borșa, as well as the Aurul settling pond in Tăuții de Sus. In contrast, the Bozânta Mare settling pond and the European Road (E 58) had a considerably lower impact on the overall Cu concentration in honey.

The results obtained for the Cu concentration aligned well with findings reported by Bartha et al. (2020) [5], who conducted research on polyfloral honey from the Copșa Mică area in Romania, with recorded values ranging from 2.00 to 33.00 mg/kg. Berinde et al. (2013) [61] also reported comparable results in their study on polyfloral honey from the nearby area of Baia Mare, Romania, with concentrations ranging from 0.20 to 0.32 mg/kg. In contrast, the results reported by Mititelu et al. (2022) [62], who studied polyfloral honey from an industrially active area with nearby refineries, and those presented by Pătruică et al. (2022) [63] for honey from the Caransebeș area, Romania, were significantly higher than those observed in the current research. Specifically, Mititelu et al. reported a value of 1.134 mg/kg, and Pătruică et al. reported values of 6.986 mg/kg for acacia honey, 5.056 mg/kg for polyfloral honey 1, and 3.947 mg/kg for polyfloral honey 2. Similar results were reported by Oroian et al. (2015) [64] with Cu concentrations of 0.18 mg/kg in acacia honey and 0.24 mg/kg in polyfloral honey.

The higher average concentration of Zn was identified in various areas, including Baia Sprie (2.10 mg/kg in polyfloral honey), Baia Mare (1.57 mg/kg in acacia honey), Baia Mare (0.98 mg/kg in chestnut honey), and Tăuții Măgherăuș (0.90 mg/kg in acacia honey) (depicted in Figure 2). Zn, recognized as an antioxidant, is present in approximately 100 enzymes, making it the second most abundant transition metal in organisms after iron (Fe). Notably, it is the sole transition metal found in all classes of enzymes, including oxidoreductases, transferases, hydrolases, lyases, isomerases, and ligases [60]. The strategic positioning of beehives near the Tăuții de Sus tailings pond (averaging 0.91 mg/kg) and Bozânta Mare (0.90 mg/kg), as well as in proximity to the Nistru, Băița, Herja mines, or the mine flotation in Băile Borșa (averaging 1.01 mg/kg), significantly influenced the accumulation of Zn in the honey samples from these regions. For instances such as chestnut honey from Baia Mare, polyfloral honey from Tăuții Măgherăuș, Baia Sprie, and acacia honey from Baia Mare, the Zn levels surpassed the maximum permissible limit set by international regulations (1.00 mg/kg). Consequently, relocating the beehives from areas close to primary sources of heavy metal pollution is recommended. It is important to consider that Zn levels in honey might be contingent upon the types of flowers bees forage on. One plausible explanation for this observation is Zn’s tendency to accumulate within biological systems [60]. The outcomes regarding the Zn concentration were in line with the findings of Berinde et al. (2013) [61] (ranging from 0.89 to 1.39 mg/kg). In contrast, the results reported by Bartha et al. (2020) [5] (15.00–36.40 mg/kg), Mititelu et al. (2022) [62] (3.886 mg/kg), Pătruică et al. (2022) [63] (4.550 mg/kg for acacia honey, 4.356 mg/kg for polyfloral honey 1, and 2.783 mg/kg for polyfloral honey 2), and Oroian et al. (2015) [64] (2.42 mg/kg for acacia honey and 3.24 mg/kg for polyfloral honey) were notably higher than the findings presented in this study.

Based on these findings, the highest average concentrations of Pb and Cd were determined to be 0.65 mg/kg and 0.02 mg/kg, respectively. In both cases, these heavy metals exhibited their highest values in acacia honey, suggesting a substantial influence of heavy metal pollution in these areas on the accumulation within honey samples (illustrated in Figure 2). In light of the Pb and Cd concentrations compared against national and international regulations, these heavy metals surpassed the permitted maximum levels for Pb (0.20 mg/kg) and Cd (0.02 mg/kg). As observed with other analyzed heavy metals, due to the elevated levels of Pb and Cd, it is advisable to relocate beehives from regions impacted by heavy metal pollution. Pb and Cd are non-essential elements and pose significant hazards to living organisms.

**Table 2 foods-12-03577-t002:** Illustrates the concentration of heavy metals in honey collected from the studied region (mg/kg WW) (mean ± standard deviation) (*n = 3*).

Areas Sample Code Year of Harvest	Distance from the Source of Pollution (~) km of the Hives	Honey Details	Denominati- On	Environment	Cu M.A.L.	Zn M.A.L.	Pb M.A.L.	Cd M.A.L.	Ni M.A.L.	Co M.A.L.	As M.A.L.	Cr M.A.L.	Hg M.A.L.
Maximum permissible levels (M.P.L)					0.50 mg/kg	1.00 mg/kg	0.20 mg/kg	0.02 mg/kg	–	–	–	–	–
Honey samples exposed to anthropogenic sources of heavy metal pollution													
Tăuții de Sus	3.5 km to the settling ponds Aurul	Raw artisan honey	Chestnut honey	Semi- rural	Near (~) 3.5 km to the settling ponds Aurul from Tăuții de Sus								
H_1-2019_ 2019		The beehives were positioned approximately (~) 3.5 km from the settling pond Aurul from Tăuții de Sus											
					0.38 ± 0.06 ^b,c,d,e^	0.16 ± 0.08 ^e,f^	0.07 ± 0.02 ^f,g^	BLD	0.12 ± 0.02 ^a,b^	BLD	BLD	1.41 ± 0.16 ^a,b^	BLD
H_1-2020_ 2020		The beehives were positioned approximately (~) 4.0 km from the settling pond Aurul from Tăuții de Sus											
					0.29 ± 0.05 ^c,d,e,f^	0.15 ± 0.05 ^e,f^	0.04 ± 0.01 ^f,g^	BLD	0.08 ± 0.06 ^c^	BLD	BLD	1.17 ± 0.17 ^a,b,c,d^	BLD
H_1-2021_ 2021		The beehives were positioned approximately (~) 3.8 km from the settling pond Aurul from Tăuții de Sus											
					0.30 ± 0.15 ^c,d,e,f^	0.19 ± 0.05 ^e,f^	0.06 ± 0.02 ^f,g^	BLD	BLD	BLD	BLD	1.65 ± 0.12 ^a^	BLD
Tăuții Măgherăuș	9.5 km to the Nistru mine and 6.0 km to the Băița mine	Raw artisan honey	Chestnut honey	Rural	Near (~) 9.5 km to the Nistru mine and 6.0 km to the Băița mine								
H_2-2018_ 2018		The beehives were positioned approximately (~) 9.0 km to the Nistru mine and 6.0 km to the Băița mine											
					0.25 ± 0.17 ^e,f^	0.71 ± 0.14 ^c,d^	0.11 ± 0.02 ^c,d,e,f^	BLD	0.02 ± 0.01 ^d^	BLD	BLD	1.31 ± 0.48 ^a,b,c^	BLD
H_2-2019_ 2019		The beehives were positioned approximately (~) 11.0 km to the Nistru mine and 8.0 km to the Băița mine											
					0.28 ±0.05 ^d,e,f^	0.83 ± 0.05 ^c^	0.09 ± 0.04 ^e,f^	BLD	BLD	BLD	BLD	1.14 ± 0.14 ^a,b,c,d^	BLD
H_2-2021_ 2021		The beehives were positioned approximately (~) 11.2 km to the Nistru mine and 8.3 km to the Băița mine											
					0.30 ± 0.08 ^c,d,e,f^	0.58 ± 0.14 ^c,d,e^	0.09 ± 0.02 ^e,f^	BLD	BLD	BLD	BLD	1.00 ± 0.03 ^b,c,d^	BLD
Baia Mare	8.0 km to the settling ponds Aurul	Raw artisan honey	Chestnut honey	Semi- rural	Near (~) 8.0 km to the settling ponds Aurul from Tăuții de Sus								
H_3-2019_ 2019		The beehives were positioned approximately (~) 8.0 km from the settling pond Aurul from Tăuții de Sus											
					0.49 ± 0.02 ^b,c^	0.73 ± 0.25 ^c,d^	0.16 ± 0.02 ^c,d,e^	0.02 ± 0.02 ^b,c^	0.13 ± 0.02 ^a,b^	BLD	BLD	1.09 ± 0.89 ^b,c,d^	BLD
H_3-2020_ 2020		The beehives were positioned approximately (~) 8.0 km from the settling pond Aurul from Tăuții de Sus											
					0.40 ± 0.05 ^b,c,d,e^	0.89 ± 0.09 ^c^	0.16 ± 0.02 ^c,d^	0.02± 0.01 ^b,c^	0.12 ± 0.01 ^a,b^	BLD	BLD	0.75 ± 0.65 ^c,d,e^	BLD
H_3-2021_ 2021		The beehives were positioned approximately (~) 8.0 km from the settling pond Aurul from Tăuții de Sus											
					0.42 ± 0.02 ^b,c,d,e^	1.31 ± 0.48 ^b^	0.10 ± 0.04 ^d,e,f^	BLD	0.11 ± 0.06 ^b^	BLD	BLD	1.18 ± 0.24 ^a,b,c,d^	BLD
Tăuții Măgherăuș	6.5 km to the settling ponds Aurul	Artisan honey	Polyfloral honey	Semi-rural	Near (~) 6.5 km to the settling ponds from Bozânta Mare								
S_4-2019_ 2019		The beehives were positioned approximately (~) 6.5 km from the settling pond Aurul from Bozânta Mare											
					0.12 ± 0.02 ^f,g^	0.90 ± 0.33 ^c^	BLD	0.01 ± 0.01 ^c^	BLD	BLD	BLD	0.28 ± 0.21 ^e,f,g^	BLD
Baia Sprie	8.0 km to the Herja mine	Artisan honey	Polyfloral honey	Rural	Near (~) 8.0 km to the Herja mine								
S_5-2019_ 2019		The beehives were positioned approximately (~) 8.0 km from the Herja mine											
					0.79 ± 0.32 ^a^	2.10 ± 0.24 ^a^	0.17 ± 0.02 ^c^	0.05 ± 0.01 ^a^	0.15 ± 0.04 ^a^	BLD	BLD	0.66 ± 0.18 ^d,e,f^	BLD
Baia Mare	8.0 km to the settling ponds Aurul	Artisan honey	Acacia	Semi- rural	Near (~) 8.0 km to the settling ponds Aurul from Tăuții de Sus								
S_6-2020_ 2020		The beehives were positioned approximately (~) 8.0 km from the settling pond Aurul from Tăuții de Sus											
					0.56 ± 0.09 ^b^	1.50 ± 0.63 ^b^	0.74 ± 0.17 ^a^	0.02 ± 0.01 ^b,c^	BLD	BLD	BLD	0.13 ± 0.02 ^f,g^	BLD
S_6-2021_ 2021		The beehives were positioned approximately (~) 8.0 km from the settling pond Aurul from Tăuții de Sus											
					0.47 ± 0.08 ^b,c,d^	1.64 ± 0.45 ^b^	0.55 ± 0.01 ^b^	0.03 ± 0.01 ^b,c^	BLD	BLD	BLD	0.16 ± 0.05 ^f,g,h^	BLD
Baia Borșa	5.1 km to the Băile Borșa mining flotation	Artisan honey	Polyfloral honey	Rural	Near (~) 5.1 km to the Băile Borșa mining flotation								
S7_-2017_ 2017		Near (~) 5.1 km to the Băile Borșa mining flotation											
					0.22 ± 0.11 ^e,f^	0.15 ± 0.05 ^e,f^	BLD	BLD	BLD	BLD	BLD	BLD	BLD
S7_-2019_ 2019		Near (~) 7.3 km to the Băile Borșa mining flotation											
					0.21 ± 0.17 ^e,f^	0.12 ± 0.04 ^f^	BLD	BLD	BLD	BLD	BLD	BLD	BLD
S7_-2020_ 2020		Near (~) 9.0 km to the Băile Borșa mining flotation											
					0.29 ± 0.14 ^c,d,e,f^	0.23 ± 0.19 ^e,f^	BLD	BLD	BLD	BLD	BLD	BLD	BLD
S7_-2021_ 2021		Near (~) 8.0 km to the Băile Borșa mining flotation											
					0.25 ± 0.14 ^e,f^	0.34 ± 0.15 ^d,e,f^	BLD	BLD	BLD	BLD	BLD	0.13 ± 0.04 ^f,g^	BLD
Satulung	5.0 km to the European Road (E 58)	Artisan honey	Polyfloral honey	Semi-rural	Near (~) 5.0 km to the European Road (E 58) with intense traffic of vehicles								
S8_-2021_ 2021		Near (~) 5.0 km to the European Road (E 58) with intense traffic of vehicles											
					0.11 ± 0.03 ^f,g^	0.10 ± 0.05 ^f^	BLD	BLD	BLD	BLD	BLD	BLD	BLD
Săcălășeni	8.0 km to the European Road (E 58)	Artisan honey	Polyfloral honey	Semi-rural	Near (~) 8.0 km to the European Road (E 58) with intense traffic of vehicles								
S9_-2020_ 2020		Near (~) 8.0 km to the European Road (E 58) with intense traffic of vehicles											
					0.28 ± 0.06 ^d,e,f^	0.15 ± 0.02 ^f^	BLD	BLD	BLD	BLD	BLD	0.12 ± 0.01 ^f,g^	BLD
Background honey samples													
Groșii Țibleșului	–	Artisan honey	Polyfloral honey	Rural	–								
S10_-2020_ 2020		–											
					0.10 ± 0.01 ^f,g^	0.17 ± 0.06 ^e,f^	BLD	BLD	BLD	BLD	BLD	BLD	BLD
Vișeu de Sus	–	Artisan honey	Polyfloral honey	-	–								
S11_-2018_ 2018				Rural	–								
					BLD	0.16 ± 0.01 ^e,f^	BLD	BLD	BLD	BLD	BLD	BLD	BLD
S11_-2019_ 2019				Semi-rural	–								
					BLD	0.14 ± 0.03 ^f^	BLD	BLD	BLD	BLD	BLD	BLD	BLD
S11_-2020_ 2020				Urban	–								
					BLD	0.15 ± 0.02 ^f^	BLD	BLD	BLD	BLD	BLD	BLD	BLD
S11_-2021_ 2021				Non-urban area	–								
					BLD	0.13 ± 0.02 ^f^	BLD	BLD	BLD	BLD	BLD	BLD	BLD
Sig.					***	***	***	***	***	–	–	***	–
Honey samples exposed to anthropogenic sources of heavy metal pollution													
Bartha et al. (2020) [5]			Polyfloral honey		2.00–33.00	15.00–36.40	0.76–3.41	0.05–3.81	–	–	–	–	–
Berinde et al. (2013) [61]			Polyfloral honey		0.20–0.32	0.89–1.39	0.12–20.34	0.076–0.093	–	–	–	–	–
Mititelu et al. (2022) [62]			Multifloral honey		1.134	3.886	0.539	0.030	0.485	–	–	1.869	–
Background honey samples													
Pătruică et al. (2022) [63]			Acacia honey		6.986 ± 0.001	4.550 ± 0.0001	0.109 ± 0.010	0.078 ± 0.001	0.249 ± 0.001	–	–	0.114 ± 0.001	–
			Polyfloral honey 1		5.056 ± 0.001	4.356 ± 0.0001	0.149 ± 0.010	0.068 ± 0.001	0.171 ± 0.001	–	–	0.106 ± 0.001	–
			Polyfloral honey 2		3.947 ± 0.001	2.783 ± 0.0001	0.097 ± 0.010	0.108 ± 0.001	0.129 ± 0.001	–	–	0.107 ± 0.001	–
Oroian et al. (2015) [64]			Acacia honey		0.18223	2.4216	0.0623	0.00114	0.1909	–	0.00864	0.0514	0.00089
			Polyfloral honey		0.23903	3.2413	0.0403	0.00263	0.1831	–	0.00559	0.0367	0.00075

Mean value ± standard deviation (n = 3). WW refers to wet weight. Letters denote significant differences (*p* ≤ 0.005) irrespective of the collection area and year. Commission Regulation (EC) No 1881/2006 dated 19 December 2006 establishing maximum levels for specific contaminants in food products. Off. J. Eur. Union 2006, L364/5–L364/24; Codex Alimentarius. Codex Alimentarius Standard for Honey 12–1981. Revised Codex Standard for Honey. Standards and Standard Methods (Vol. 11). 2001; Council Directive 2001/110/EC Regarding Honey. EU Off. J. 2002, L10, 47–52. BLD stands for below the detection limit (LoQ): LoQ for Cu: 1.545 µg/L, LoQ for Pb: 0.231 µg/L, LoQ for Cd: 0.069 µg/L, LoQ for Co: 0.136 µg/L, LoQ for As: 0.743 µg/L; LoQ for Hg: 0.1379 µg/L. Throughout the years of sample collection, minor adjustments were observed in relation to pollution sources, facilitated by remote hive relocation as an effective measure. *** = There are significant differences between the analyzed samples.

### 3.3. Correlation Matrix Depicting the Relationships between Key Heavy Metals Found in Honey and the Primary Factors That Can Impact Their Accumulation

Table 3 displays the Spearman’s correlation among the various assessed toxic elements and factors such as region (A.), proximity to pollution source (DpS.), production year of honey (Y.), and honey type (Ht).

Noteworthy negative correlations were also established between the Pb/honey type (r^2^ = −0.413 *) and the Ni/honey type (r^2^ = −0.454 *). Additionally, it was observed that the concentration of Cr in honey was significantly impacted by all the aforementioned factors considered in the study, with the interaction of Zn also playing a role (r^2^ = 0.478 *).

The most pronounced positive correlations were observed between Zn/Cu (r^2^ = 0.678 *), Pb/Cu (r^2^ = 0.555 *), Pb/Zn (r^2^ = 0.629 *), Cd/Cu (r^2^ = 0.588 *), Cd/Zn (r^2^ = 0.503 *), Ni/Cu (r^2^ = 0.557 *), and Ni/Cd (r^2^ = 0.461 *). Through Spearman’s correlation analysis, certain interactions among the toxic elements examined in honey were identified, indicating a direct impact on their accumulation within the honey. The most significant negative correlation was observed between Cr/Cu (r^2^ = −0.511 *). It is important to note that the correlation between toxic elements and honey can be influenced by various factors, including the geographical location, environmental conditions, and sources of contamination. Furthermore, notable positive and negative associations were uncovered between the hive location and area, year/type of honey, and proximity to primary pollution sources. Specifically, Cu/area exhibited a strong negative correlation (r^2^ = −0.864 **), while Cu/distance from the pollution source showed a similarly robust negative correlation (r^2^ = −0.459 **).

It is worth emphasizing that while this discovery lends support to the hypothesis, further investigations and in-depth data analysis might be necessary to substantiate the relationship and gain a nuanced comprehension of the specific underlying mechanisms. Additionally, it is crucial to take into account other variables such as local agricultural practices, geological variations, and potential alternative sources of pollution. This all-encompassing approach is crucial to gaining a complete understanding of the contamination of toxic elements in honey. It is imperative to consider these factors as they can substantially impact the concentrations of toxic elements and should be factored in when assessing both the origins and potential routes of contamination.

### 3.4. Evoluation of Bioaccumulation Factor for Heavy Metals in Honey

An additional objective of this study was to establish a comprehensive overview of heavy metal contamination levels through the application of a bioaccumulation factor. The bioaccumulation factor (BFA) is a quantitative measure that depicts the accumulation of specific substances, such as heavy metals, within the tissues of an organism in relation to the concentration of the same substance in the surrounding environment, typically water or soil [65]. This parameter proves valuable in assessing an organism’s capacity to gather particular substances from its surroundings [65]. Bioaccumulation factors play a pivotal role in environmental investigations, offering insights into the potential risks associated with certain substances, particularly hazardous pollutants like heavy metals [66]. They aid in comprehending the upward movement of these substances in the food chain, potentially endangering higher trophic-level organisms, including humans, due to biomagnification effects. Vigilant monitoring of bioaccumulation factors holds significance in the management of environmental pollution and the preservation of both ecosystem integrity and human well-being [66]. The bioaccumulation factor was computed by dividing the concentration of a specific heavy metal in honey by its concentration in the soil [66].

The calculated outcomes of the bioaccumulation factor (BAF), as depicted in Table 4, revealed that the sequential transfer of metals from the soil to honey followed the sequence Cr > Ni > Cd > Zn > Pb > Cu. When considering the honey type and the production area, the BAF results indicated that the metal transfer from soil to honey exhibited the sequence H6 > H5 > H1 > H2 > H4 > H3 > H7. Remarkably, regardless of the honey type or production area, the accumulation pattern of Cr raised a notable concern. Notably, the research area under investigation displayed indications of heavy metal pollution, particularly with Cd, Cu, Pb, and Zn [67], whereas no prior studies have reported Cr pollution in this specific area. As a plausible explanation for the observed Cr accumulation behavior, we hypothesized that it could be linked to the application of phytosanitary treatments in nearby agricultural activities, which might have reached the vicinity of the bee’s habitat.

## 4. Conclusions

The study analyzed 38 soil samples from the Maramureș region, focusing on nine potentially toxic elements at a 0–10 cm depth. The highest concentrations were copper (Cu) at 3286.65 mg/kg, zinc (Zn) at 2834.58 mg/kg, lead (Pb) at 1205.57 mg/kg, chromium (Cr) at 10.75 mg/kg, nickel (Ni) at 7.99 mg/kg, cadmium (Cd) at 6.33 mg/kg, cobalt (Co) at 5.98 mg/kg, arsenic (As) at 4.37 mg/kg, and mercury (Hg) below the detection limit. Soil samples near anthropogenic areas, like mining operations and settling ponds, had significantly higher metal concentrations, with Aurul settling pond and Herja mine being major sources. Copper and zinc exceeded legal limits in some areas, posing environmental risks. Acacia honey had the highest copper levels, influenced by nearby mining sites. Lead and cadmium concentrations exceeded legal limits in certain areas, likely due to the Aurul settling pond’s influence. The study emphasizes stricter adherence to environmental regulations.

The research also examined heavy metal concentrations in different honey types (chestnut, acacia, and polyfloral) from various locations. Chromium had the highest mean concentration at 0.58 mg/kg, followed by zinc (0.56 mg/kg), copper (0.34 mg/kg), lead (0.10 mg/kg), nickel (0.03 mg/kg), cadmium (0.01 mg/kg), and cobalt below the detection limit. Arsenic and mercury were undetected. Copper concentrations in honey were notably high in the Baia Mare area, mainly due to former mining sites and settling ponds. Zinc levels exceeded international limits in some areas, warranting hive relocation. Acacia honey had the highest lead and cadmium concentrations, surpassing permissible levels. The study highlights the importance of monitoring heavy metal contamination in honey for human and environmental safety.

The analysis found significant correlations between toxic elements in honey, with notable positive correlations for Zn/Cu, Pb/Cu, Pb/Zn, Cd/Cu, Cd/Zn, Ni/Cu, and Ni/Cd. A negative correlation was identified between Cr/Cu. The hive location and area, year/type of honey, and proximity to pollution sources also influenced metal concentrations in honey. Cu/area had a strong negative correlation. Pb/honey type and Ni/honey type showed negative correlations. Cr concentrations were influenced by these factors and Zn interaction. Further investigations are needed to validate these relationships, considering local agricultural practices, geological variations, and alternative pollution sources. The study employed the bioaccumulation factor (BAF) to assess metal transfer from soil to honey. The sequential transfer of metals from soil to honey was Cr > Ni > Cd > Zn > Pb > Cu. BAF analysis by honey type and production area revealed variation in metal accumulation patterns, with Cr accumulation raising concerns. This region showed signs of heavy metal pollution, particularly Cd, Cu, Pb, and Zn. Unexpectedly, Cr pollution was observed, possibly due to nearby agricultural phytosanitary treatments.

## Figures and Tables

**Figure 1 foods-12-03577-f001:**
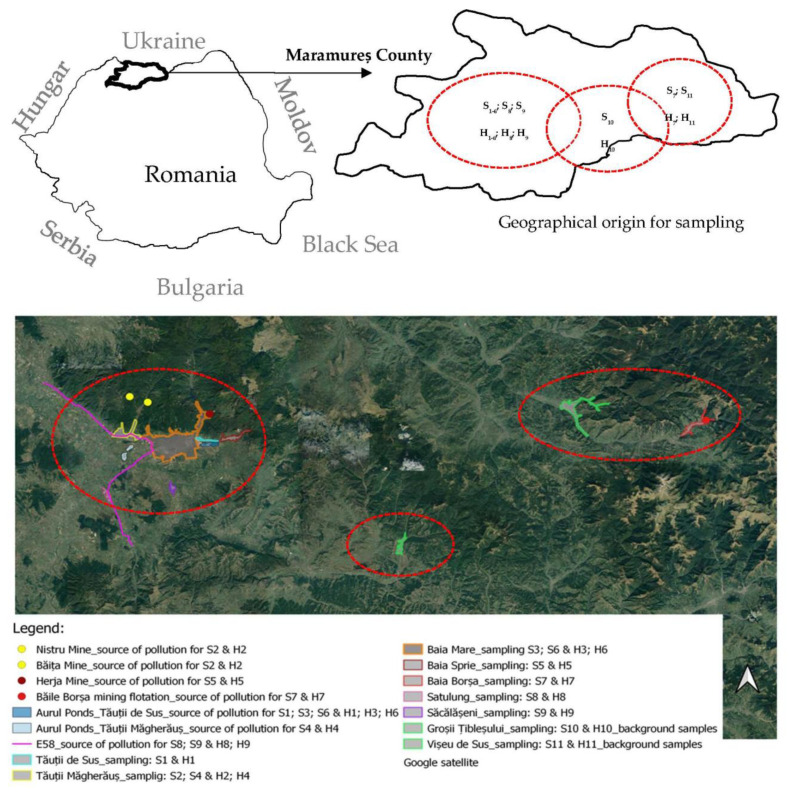
Geographical sources of the soil and honey samples collected. Each designated code for both soil and honey samples corresponds to the identified pollution source for the respective samples.

**Figure 2 foods-12-03577-f002:**
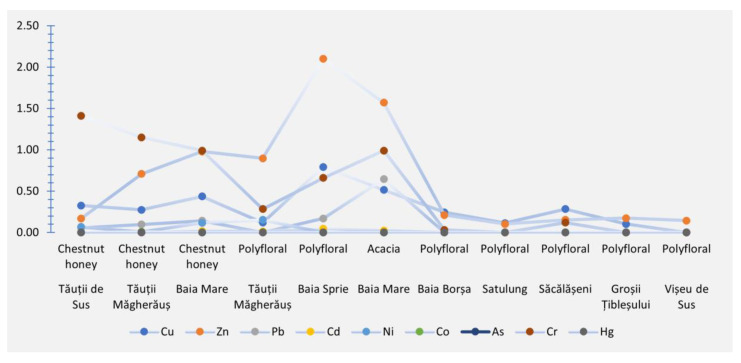
The dispersion of heavy metal concentrations (*x* axis) (mg/kg) based on the geographical origin (*y* axis) of the honey samples irrespective of the specific collection year.

**Table 1 foods-12-03577-t001:** The concentration of heavy metals in soil from studied areas (measured in mg/kg dry weight) (mean ± standard deviation) (*n = 3*).

**Areas** **Sample Code** **Year of Harvest**	**Distance from the Source of Pollution (~) km**	**Sampling** **Depth** **(0–20 cm)**	**Cu** **M.A.L. ***	**Zn** **M.A.L. ***	**Pb** **M.A.L. ***	**Cd** **M.A.L. ***	**Ni** **M.A.L. ***	**Co** **M.A.L. ***	**As** **M.A.L. ***	**Cr** **M.A.L. ***	**Hg** **M.A.L. ***
			20 mg/kg	100 mg/kg	20 mg/kg	1 mg/kg	20 mg/kg	15 mg/kg	5 mg/kg	30 mg/kg	0.1 mg/kg
**Alert Threshold**		**Susceptible**	100 mg/kg	300 mg/kg	50 mg/kg	3 mg/kg	75 mg/kg	30 mg/kg	15 mg/kg	100 mg/kg	1 mg/kg
		**Less Susceptible**	250 mg/kg	700 mg/kg	250 mg/kg	5 mg/kg	200 mg/kg	100 mg/kg	25 mg/kg	300 mg/kg	4 mg/kg
**Intervention Threshold**		**Susceptible**	200 mg/kg	600 mg/kg	100 mg/kg	5 mg/kg	150 mg/kg	50 mg/kg	25 mg/kg	300 mg/kg	2 mg/kg
		**Less susceptible**	500 mg/kg	1500 mg/kg	1000 mg/kg	10 mg/kg	500 mg/kg	250 mg/kg	50 mg/kg	600 mg/kg	10 mg/kg
Soil samples exposed to anthropogenic sources of heavy metal pollution											
Tăuții de Sus	Near (~) 3.5 km to the settling ponds Aurul from Tăuții de Sus										
S_1-2019_ 2019	The beehives were positioned approximately (~) 3.5 km from the settling pond Aurul from Tăuții de Sus										
			156.97 ± 47.60 ^i,î^	1422.58 ± 73.90 ^e,f^	54.96 ± 12.68 ^c,d,e^	BLD	2.23 ± 0.35 ^d,e^	0.37 ± 0.19 ^d^	0.59 ± 0.31 ^c^	7.52 ± 0.96 ^b^	BLD
S_1-2020_ 2020	The beehives were positioned approximately (~) 4.0 km from the settling pond Aurul from Tăuții de Sus										
			177.48 ± 15.40 ^h,i^	1580.15 ± 111.73 ^d,e^	62.81 ± 16.57 ^c,d,e^	BLD	1.93 ± 1.14 ^d,e,f^	0.25 ± 0.01 ^d^	0.64 ± 0.10 ^c^	7.73 ± 1.39 ^b^	BLD
S_1-2021_ 2021	The beehives were positioned approximately (~) 3.8 km from the settling pond Aurul from Tăuții de Sus										
			184.89 ± 13.50 ^h,i^	1435.80 ± 178.24 ^e,f^	50.96 ± 21.88 ^c,d,e^	BLD	1.75 ± 0.24 ^d,e,f^	0.31 ± 0.06 ^d^	0.38 ± 0.10 ^c^	6.12 ± 0.86 ^b^	BLD
Tăuții Măgherăuș	Near (~) 9.5 km to the Nistru mine and 6.0 km to the Băița mine										
S_2-2018_ 2018	The beehives were positioned approximately (~) 9.0 km to the Nistru mine and 6.0 km to the Băița mine										
			252.43 ± 113.10 ^g,h,i^	2588.59 ± 390.84 ^b^	107.72 ± 7.72 ^c^	0.03 ± 0.01 ^d^	1.06 ± 0.15 ^e,f^	0.32 ± 0.15 ^d^	0.13 ± 0.02 ^c^	6.31 ± 0.73 ^b^	BLD
S_2-2019_ 2019	The beehives were positioned approximately (~) 11.0 km to the Nistru mine and 8.0 km to the Băița mine										
			361.18 ± 52.82 ^g^	2834.58 ± 46.06 ^a^	104.23 ± 8.89 ^c^	0.02 ± 0.01 ^d^	1.14 ± 0.14 ^e,f^	0.29 ± 0.23 ^d^	0.17 ± 0.07 ^c^	6.83 ± 0.54 ^b^	BLD
S_2-2021_ 2021	The beehives were positioned approximately (~) 11.2 km to the Nistru mine and 8.3 km to the Băița mine										
			327.83 ± 51.67 ^g,h^	1892.02 ± 220.06 ^c^	101.52 ± 13.57 ^c^	BLD	0.87 ± 0.12 ^e,f^	0.48 ± 0.16 ^d^	0.17 ± 0.02 ^c^	6.52 ± 0.16 ^b^	BLD
Baia Mare	Near (~) 8.0 km to the settling ponds Aurul from Tăuții de Sus										
S_3-2019_ 2019	The beehives were positioned approximately (~) 8.0 km from the settling pond Aurul from Tăuții de Sus										
			2856.86 ± 246.90 ^c^	1738.68 ± 118.75 ^c,d^	596.52 ± 52.83 ^b^	2.19 ± 0.57 ^c^	6.78 ± 1.15 ^a,b^	2.18 ± 0.45 ^b^	4.09 ± 0.50 ^a^	10.07 ± 1.89 ^a^	BLD
S_3-2020_ 2020	The beehives were positioned approximately (~) 8.0 km from the settling pond Aurul from Tăuții de Sus										
			3286.65 ±143.82 ^a^	1744.15 ± 102.07 ^c,d^	605.36 ± 112.46 ^b^	2.65 ± 0.53 ^c^	6.55 ± 0.66 ^a,b^	1.97 ± 1.46 ^b,c^	4.37 ± 1.85 ^a^	10.71 ± 2.83 ^a^	BLD
S_3-2021_ 2021	The beehives were positioned approximately (~) 8.0 km from the settling pond Aurul from Tăuții de Sus										
			3123.24 ± 135.22 ^b^	1712.23 ± 58.15 ^c,d^	594.93 ± 51.39 ^b^	2.48 ± 0.30 ^c^	6.66 ± 0.74 ^a,b^	1.59 ± 0.38 ^b,c^	3.60 ± 0.10 ^a^	10.75 ± 3.01 ^a^	BLD
Tăuții Măgherăuș	Near (~) 6.5 km to the settling ponds from Bozânta Mare										
S_4-2019_ 2019	The beehives were positioned approximately (~) 6.5 km from the settling pond Aurul from Bozânta Mare										
			565.99 ± 77.85 ^f^	770.72 ± 65.58 ^g^	23.39 ± 8.63 ^d,e^	0.97 ± 0.36 ^d^	3.19 ± 1.47 ^c,d^	BLD	1.80 ± 1.19 ^b^	1.87 ± 1.20 ^c,d,e^	BLD
Baia Sprie	Near (~) 8.0 km to the Herja mine										
S_5-2019_ 2019	The beehives were positioned approximately (~) 8.0 km from the Herja mine										
			2351.35 ± 97.01 ^d^	1317.23 ± 136.89 ^f^	82.87 ± 7.54 ^c,d^	4.41 ± 0.93 ^b^	7.99 ± 1.46 ^a^	0.94 ± 0.08 ^c,d^	0.76 ± 0.28 ^c^	3.05 ± 0.18 ^c^	BLD
Baia Mare	Near (~) 8.0 km to the settling ponds Aurul from Tăuții de Sus										
S_6-2020_ 2020	The beehives were positioned approximately (~) 8.0 km from the settling pond Aurul from Tăuții de Sus										
			1171.83 ± 142.11 ^e^	2533.76 ± 117.59 ^b^	1205.57 ± 70.56 ^a^	6.33 ± 1.57 ^a^	3.85 ± 1.53 ^c^	5.98 ± 1.67 ^a^	2.10 ± 0.50 ^b^	0.79 ± 0.08 ^d,e^	BLD
S_6-2021_ 2021	The beehives were positioned approximately (~) 8.0 km from the settling pond Aurul from Tăuții de Sus										
			1809.32 ± 52.73 ^e^	2379.58 ± 403.63 ^b^	1166.11 ± 68.21 ^a^	5.52 ± 1.90 ^a^	5.96 ± 2.11 ^b^	5.64 ± 2.09 ^a^	2.60 ± 0.96 ^b^	0.68 ± 0.38 ^d,e^	BLD
Baia Borșa	Near (~) 5.1 km to the Băile Borșa mining flotation										
S7_-2017_ 2017	Near (~) 5.1 km to the Băile Borșa mining flotation										
			361.24 ± 34.85 ^g^	118.73 ± 18.36 ^h^	62.09 ± 14.86 ^c,d,e^	BLD	0.89 ± 0.34 ^e,f^	BLD	BLD	1.11 ± 0.10 ^c,d,e^	BLD
S7_-2019_ 2019	Near (~) 7.3 km to the Băile Borșa mining flotation										
			374.46 ± 32.72 ^g^	90.39 ± 14.05 ^h^	65.94 ± 11.29 ^c,d,e^	BLD	1.60 ± 0.84 ^e,f^	BLD	BLD	1.30 ± 0.34 ^c,d,e^	BLD
S7_-2020_ 2020	Near (~) 9.0 km to the Băile Borșa mining flotation										
			320.18 ± 5.13 ^g,h^	80.02 ± 3.62 ^h^	45.75 ± 8.84 ^c,d,e^	BLD	0.59 ± 0.06 ^e,f^	BLD	BLD	1.47 ± 0.22 ^c,d,e^	BLD
S7_-2021_ 2021	Near (~) 8.0 km to the Băile Borșa mining flotation										
			369.53 ± 52.19 ^g^	83.05 ± 1.43 ^h^	71.64 ± 13.91 ^c,d^	BLD	0.57 ± 0.26 ^e,f^	BLD	BLD	0.95 ± 0.29 ^d,e^	BLD
Satulung		Near (~) 5.0 km to the European Road (E 58) with intense traffic of vehicles									
S8_-2021_ 2021	Near (~) 5.0 km to the European Road (E 58) with intense traffic of vehicles										
			3.39 ± 1.28 ^î^	12.56 ± 2.24 ^h^	3.59 ± 1.69 ^e^	BLD	0.42 ± 0.23 ^f^	BLD	0.04 ± 0.03 ^c^	0.72 ± 0.43 ^d,e^	BLD
Săcălășeni		Near (~) 8.0 km to the European Road (E 58) with intense traffic of vehicles									
S9_-2020_ 2020	Near (~) 8.0 km to the European Road (E 58) with intense traffic of vehicles										
			0.78 ± 0.51 ^î^	15.18 ± 5.52 ^h^	2.27 ± 1.05 ^e^	BLD	1.32 ± 0.48 ^e,f^	BLD	BLD	0.53 ± 0.13 ^d,e^	BLD
Background soil samples											
Groșii Țibleșului	–										
S10_-2020_ 2020											
			1.21 ± 0.01 ^î^	42.95 ± 9.48 ^h^	0.02 ± 0.01 ^e^	BLD	2.11 ± 0.51 ^d,e^	BLD	BLD	2.33 ± 0.29 ^c,d^	BLD
Vișeu de Sus	–										
S11_-2018_ 2018	–										
			2.74 ± 1.05 ^î^	34.35 ± 6.89 ^h^	BLD	BLD	1.02 ± 0.08 ^e,f^	BLD	BLD	0.13 ± 0.03 ^e^	BLD
S11_-2019_ 2019	–										
			3.80 ± 1.14 ^î^	23.81 ± 6.72 ^h^	0.02 ± 0.01 ^e^	BLD	1.03 ± 0.43 ^e,f^	BLD	BLD	0.13 ± 0.02 ^e^	BLD
S11_-2020_ 2020	–										
			1.52 ± 0.06 ^î^	35.52 ± 11.40 ^h^	BLD	BLD	1.11 ± 0.38 ^e,f^	BLD	BLD	0.15 ± 0.06 ^e^	BLD
S11_-2021_ 2021	–										
			0.99 ± 0.23 ^î^	34.40 ± 3.63 ^h^	BLD	BLD	1.28 ± 0.36 ^e,f^	BLD	BLD	0.13 ± 0.02 ^e^	BLD
Sig.			***	***	***	***	***	***	***	***	–
Soil samples exposed to anthropogenic sources of heavy metal pollution											
Huzum et al. (2012) [54]			256.00	60.10	12.90	0.21	29.90	7.20	11.20	208.00	–
Bora et al. (2020) [50]			621.79–4155.95	45.36–3483.25	6.62–4262.23	0.12–32.53	6.97–28.60	5.08–29.57	1.15–5.13	2.72 ± 0.65	0.034–0.070
Chakraborty et al. (2017) [55]			19.8–2760.0	54.4–2370.0	38.0–14,329.0	–	–	–	7.8–889.0	–	–
Paulette et al. (2015) [56]			77–7675	–	705–10,074	–	–	–	–	–	–
Mihali et al. (2017) [57]			40.9–621.6	82.26–1002	48.12–3472	0.04–11	4.98–9.06	3.3–8.2	0.61–80.1	–	–
Albulescu et al. (2009) [58]			36.63–112.00	–	21.90	1.77	24.55	–	–	13.32	–
Bora et al. (2015) [47]			479.64 ± 53.97	69.44 ± 4.02	14.77 ± 0.74	0.45 ± 0.10	16.28 ± 2.01	9.75 ± 1.47	–	–	–
Background soil samples											
European Communities Council (1986) [59]			50–140	150–300	50–300	1–3	30–75	–	–	–	1–1.5
Kabata-Pendias, (2010) [59]			13–24	45–100	22–44	0.37–0.78	12.0–34	–	0–9.3	–	–
Common abundance in topsoil [59]			5–50	10–100	–	0.1–1	20–50	–	0.1–55	–	–
Phytotoxic levels of elements in soils [59]			36–698	100–1.000	–	–	100	–	200	–	–

Average value along with the standard deviation (*n* = 3) is indicated. The abbreviation “DW” corresponds to dry weight. The use of letters signifies significance of differences (*p* ≤ 0.005), regardless of the area and year of sample collection. * Reference to Order of the Ministry of Waters, Forests and Environmental Protection No. 756/3 November 1997, endorsing the regulation on environmental pollution assessment in Bucharest, Romania; 1997.M.A.L. (maximum admissible limit) represents normal values. “in” denotes insignificance. “BLD” signifies values that are below the detection limit (LoQ): LoQ for Pb: 0.231 µg/L, LoQ for Cd: 0.069 µg/L, LoQ for Co: 0.136 µg/L, LoQ for As: 0.743 µg/L; LoQ for Hg: 0.1379 µg/L. Throughout the years of sample collection, minor adjustments were observed in relation to pollution sources, facilitated by remote hive relocation as an effective measure. *** = There are significant differences between the analyzed samples.

**Table 3 foods-12-03577-t003:** Spearman’s correlation matrix depicting the relationships between the primary heavy metals found in honey and the key factors that can impact their accumulation.

**Analyzed** **Parameter**	**A.**	**DpS.**	**Y.**	**Ht.**	**Ai.**	**Cu**	**Zn**	**Pb**	**Cd**	**Ni**	**Cr**	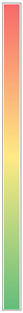	**−1**
A.	1											−0.8
DpS.		1										−0.6
Y.	-	-	1								−0.4
Ht.	-	-	-	1								−0.2
Ai.	-	-	-	-	1							0
Cu	−0.864 **	−0.459 *	−0.272	−0.056	−0.615 *	1						0.2
Zn	−0.386	−0.125	−0.117	0.177	−0.492 *	0.678 *	1					0.4
Pb	−0.333	−0.135	−0.077	0.413 *	−0.525 *	0.555 *	0.629 *	1				0.6
Cd	−0.216	−0.014	−0.257	0.227	−0.302	0.588 *	0.690 *	0.503 *	1			0.8
Ni	−0.378	−0.342	−0.381	−0.454 *	−0.465 *	0.557 *	0.364	0.079	0.461 *	1		1
Cr	−0.769 *	−0.639 *	−0.089	−0.511 *	−0.830 *	−0.511 *	0.478 *	0.353	0.371	0.214	1

A = area; DpS. = distance from the pollution source; Y. = year of honey production; Ht. = honey type; Ai = anthropogenic influence. * Significant correlation at *p* < 0.05 (95% confidence). ** Highly significant correlation at *p* < 0.01 (99% confidence); N = 72. The correlation coefficient “r” lies within the range of −1 to 1. A value of −1 signifies a perfect negative correlation, implying that one variable decreases linearly as the other variable increases. Conversely, a value of 1 indicates a perfect positive correlation, where both variables increase linearly. A value of 0 denotes no linear correlation between the variables. The strength of the relationship is indicated by the magnitude of the correlation coefficient. As the absolute value of “r” approaches 1, the correlation becomes stronger. For instance, an “r” value of −0.8 or 0.8 signifies a robust negative or positive correlation, respectively.

**Table 4 foods-12-03577-t004:** The evaluation of bioaccumulation factors for heavy metals in honey.

**Denomination**	**Area Sample**	**Cu**	**Zn**	**Pb**	**Cd**	**Ni**	**Co**	**As**	**Cr**	**Hg**	**TOTAL**
Chestnut	Tăuții de Sus (H1)	0.0019	0.0001	0.0010	0	0.0318	0	0	0.2028	0	0.2376
Chestnut	Tăuții Măgherăuș (H2)	0.0009	0.0003	0.0009	0	0.0063	0	0	0.1760	0	0.1844
Chestnut	Baia Mare (H3)	0.0001	0.0006	0.0002	0.006	0.0180	0	0	0.0960	0	0.1205
Polyfloral	Tăuții Măgherăuș (H4)	0.0002	0.0012	0	0.010	0	0	0	0.1497	0	0.1614
Polyfloral	Baia Sprie (H5)	0.0003	0.0016	0.0021	0.011	0.0188	0	0	0.2164	0	0.2505
Acacia	Baia Mare (H6)	0.0004	0.0006	0.0005	0.004	0	0	0	1.3699	0	1.3757
Polyfloral	Baia Borșa (H7)	0.0007	0.0014	0	0	0	0	0	0.0342	0	0.0363
**TOTAL**		0.0045	0.0058	0.0048	0.0315	0.0748	0	0	2.2450	0	2.3664
**Denomination**	**Area Sample**	**Heavy Metal Concentration**									
Chestnut	Tăuții de Sus (H1)			Cr > Ni > Cu > Pb > Zn							
Chestnut	Tăuții Măgherăuș (H2)			Cr > Ni > Cu = Pb > Zn							
Chestnut	Baia Mare (H3)			Cr > Ni > Zn = Cd > Pb > Cu							
Polyfloral	Tăuții Măgherăuș (H4)			Cr > Cd > Zn > Cu > Zn > Ni							
Polyfloral	Baia Sprie (H5)			Cr > Ni > Pb > Zn > Cd > Cu							
Acacia	Baia Mare (H6)			Cr > Zn > Pb > Cu > Cd							
Polyfloral	Baia Borșa (H7)			Cr > Zn > Cu							

The bioaccumulation index was computed solely for regions subjected to anthropogenic sources of heavy metal pollution. The designated areas for calculation were as follows: H1 = Tăuții de Sus/chestnut; H2 = Tăuții Măgherăuș/chestnut; H3 = Baia Mare/chestnut; H4 = Tăuții Măgherăuș/polyfloral; H5 = Baia Sprie/polyfloral; H6 = Baia Mare/acacia; H7 = Baia Borșa/acacia. In the context of Co, As, and Hg, the bioaccumulation index could not be determined due to the absence of detection of these elements within the honey samples.

## Data Availability

The data used to support the findings of this study can be made available by the corresponding author upon request.

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
