# Peer review of "Unravelling Heavy Metal Dynamics in Soil and Honey: A Case Study from Maramureș Region, Romania"

_foods, 2023, doi:10.3390/foods12193577_

Round 1
Reviewer 1 Report
A summary, focusing on the heavy metals, the authors collected a series of soil and honey samples and evaluated the concentrations to assess the bioaccumulation. Overall, the topic is interesting, the manuscript is well-organized and written.
1. In part 2.1, “A total of thirty-eight soil samples, designated as S1-S11, and thirty-nine Apis mellifera honey samples, denoted as H1-S11,”, both the soil samples and honey samples give the name of S11, which is confusing, please double-check. Also, it said thirty-nine honey samples were collected, but there are just 38 samples when adding the numbers together.
2. In part 2.1, it is said the honey samples were gathered in 2017 and 2021, however, in part 2.2.2, it said the samples were obtained from the seasons of 2017-2020; the time point is inconsistent.
3. Figures 2 and 3 are hard to read, the colors for the dots and lines are hard to discriminate; Also, please add the legend of the x- and y-axis.
4. The authors assess the heavy metals dynamics in soil and honey, if you could also evaluate the heavy metals in the related flowers, the story will be more complete.
Author Response
Reviewer 1
The suggestions of reviewer 1 were made in red to stand out.
A summary, focusing on the heavy metals, the authors collected a series of soil and honey samples and evaluated the concentrations to assess the bioaccumulation. Overall, the topic is interesting, the manuscript is well-organized and written.
- In part 2.1, “A total of thirty-eight soil samples, designated as S1-S11, and thirty-nine Apis mellifera honey samples, denoted as H1-S11,”, both the soil samples and honey samples give the name of S11, which is confusing, please double-check. Also, it said thirty-nine honey samples were collected, but there are just 38 samples when adding the numbers together.
It has been modified so that the soil samples are marked with S (S1-S11) and the honey samples are marked with H (H1-H11)...so that there is no more confusion regarding the name of the analyzed samples. Also here it was modified so that the total number of honey samples is 38.
- In part 2.1, it is said the honey samples were gathered in 2017 and 2021, however, in part 2.2.2, it said the samples were obtained from the seasons of 2017-2020; the time point is inconsistent.
It has been revised so that there is no more confusion; the samples were collected between 2017-2021
- Figures 2 and 3 are hard to read, the colours for the dots and lines are hard to discriminate; Also, please add the legend of the x- and y-axis.
The role of this image is to highlight the distribution of heavy metals in soil and honey, depending on the sample collection area, I intentionally set the image so that the image is not overloaded...but to distinguish the analyzed samples depending on the metal concentration, we recommend using the tables. In the future research, we will use a more efficient and simplified way for the graphic presentation of the results. The legend of the x and y axes have been added to both figures…in the title so that they are as visible as possible without loading the graph even more.
- The authors assess the heavy metals dynamics in soil and honey, if you could also evaluate the heavy metals in the related flowers, the story will be more complete.
Thank you very much for the suggestion sent regarding the analysis of heavy metal concentrations in flowers... in future research we will consider the suggestion received.

Reviewer 2 Report
This manuscript, “Unravelling heavy metals dynamics in soil and honey: A case study from MaramureÈ™ region, Romania”, tested the heavy metals concentration of soil and honey samples from different locations in MaramureÈ™ region.
First, based on the soil samples, it was found that, “samples near anthropogenic sources displayed elevated metal levels, with the Aurul settling pond and Herja mine being major contamination sources. Copper concentrations exceeded legal limits in areas near these sources. Zinc concentrations were highest near mining areas, and Pb and Cd levels surpassed legal limits near beehives producing acacia honey. Nickel and Co levels were generally within limits but elevated near the Herja mine.”
Second, based on the honey samples, it was found that, “Positive correlations were identified between certain elements in honey, influenced by factors like location and pollution sources.”.
The toxic element contamination in honey addressed the pollution sources and pathways, which bring forward the need for pollution control monitor to assure the honey safety.
Overall, this manuscript is well-organized, the experimental, discussions and conclusions are logical, clear, and rational. Therefore, I would like to recommend it to Foods.
The English Language is clear, it will be better if the authors can make it concise.
Author Response
Reviewer 2
The suggestions of reviewer 2 were made in red to stand out.
This manuscript, “Unravelling heavy metals dynamics in soil and honey: A case study from MaramureÈ™ region, Romania”, tested the heavy metals concentration of soil and honey samples from different locations in MaramureÈ™ region.
First, based on the soil samples, it was found that, “samples near anthropogenic sources displayed elevated metal levels, with the Aurul settling pond and Herja mine being major contamination sources. Copper concentrations exceeded legal limits in areas near these sources. Zinc concentrations were highest near mining areas, and Pb and Cd levels surpassed legal limits near beehives producing acacia honey. Nickel and Co levels were generally within limits but elevated near the Herja mine.”
Second, based on the honey samples, it was found that, “Positive correlations were identified between certain elements in honey, influenced by factors like location and pollution sources.”.
The toxic element contamination in honey addressed the pollution sources and pathways, which bring forward the need for pollution control monitor to assure the honey safety.
Overall, this manuscript is well-organized, the experimental, discussions and conclusions are logical, clear, and rational. Therefore, I would like to recommend it to Foods.
Thank you very much for the suggestions sent....personally but also with the team of researchers with whom I collaborate, we are fighting with the Romanian authorities so that more absolutely necessary strategies are implemented for this area in Romania. We hope that such investigations will raise the alarm regarding the existence of tailings dumps... which present a real time bomb for the environment and for the residents of this area.

Reviewer 3 Report
Comments
Rows 54-66 These rows can be deleted
Row 57 …..ailments
Aliments
Rows 69-75 These rows can be deleted….are too generic
Rows 79-80 These rows can be deleted ….are too generic
Rows 105-121 These rows can be deleted….. are too generic
Rows 149-152 These rows can be deleted….. are too generic
Rows 167-170 I would have verified before the chain soil-flower and successively the link soil-flower-honey….The direct link soil-honey with all the problems reported in your introduction, rows 89-97, seems too difficult to study also for the aim of rows 173-175.
Row 178 A total of thirty-eight soil samples, designated as S1-S11, and thirty-nine Apis mellifera honey samples, denoted as H1-S11
I thought S1-S38 and ……H1-H39
Rows 185-188 All the beehives were away from pollution sources and therefore were all honey’s samples of these beehives?
Rows 203-205 Did soil background (and honey?) “produce” only polyfloral honey?
Rows 214-217 Have You evidenced differences between manual or mechanical ectraction?
Rows 246-252 Therefore it is always better to treat all honey sample by breaking the organic matrix (acid treatment in microwave or at 550°C).
Rows 253-259 It would be better at the start of paragraph, before rows 246-252.
Rows 259-264 I don’t understand these rows…..in the following (264-267) you treated directly honey in microwave……
Row 269…. Basi ICP-MS Analytical Instrumental Parameters
Basic?
Row 290 …. 99.99%
Better if you write Ar 5.0 and He 6.0 or 99.999 and 99.9999…………
Table S5 Three or four decimal figures? LoQ (also LoD) and BEC values are based on different calculations: What have you considered? In the text BEC is not considered……
Row 339 … test portion of the sample…
Soil? Honey? Where are the results?
Row 342 …wine sample..
Why not honey?
Figure S1 and S2 They can be deleted…better a table
Rows 365-368 Without standard deviation values the two decimal figures don’t seem significance….i.e. Cu 3286.65…..better 3286…..
Rows 368-370 …Mercury (Hg) with concentrations below the limit of detection (BLD) which was equal to or greater than the limit of quantification (LoQ) set at 0.1379 μg/L for Hg.
What is the meaning? Limit of detection equal or greater of limit of quantification???? Normally is the reverse…..
Rows 370-374 Why not in the table 1?
Rows 381-398 Better the information of table 1….
Row 383 As above (rows 365-368)
Rows 405-6
How much?
Rows 407-419 It would be sufficient for the references a table…In the soil cultivated with vines high Cu values are typical…and the same for others metals…..
Rows 439-444 It would be sufficient for the references or a table
Figure 2 Don’t give significance informations….better a table.
Rows 468-475 and 489-494 It would be sufficient for the references a table
Rows 516-520 It would be sufficient for the references a table
In all Table 1……for example 156.97 ± 47.60 ……. 157 ± 48….the digital figures when are significance.
Row 530…. to be below the detection limit (LoQ for Hg: 0.1379 μg/L)
Therefore ….below the quantification limit….
Row 532 Table S8 ….Environment
What is the meaning of environment? How is considered?
Row 537 … varying degrees…
What is the meaning in this context? Perhaps it is better delete…
Rows 539-542 To contextualize these findings, a comparison was made with the maximum permissible contaminant levels for heavy metals in both food (as specified by Commission Regulation (EC) No 1881/2006) and honey (according to Codex Alimentarius and Council Directive 2001/110/EC).
If there is a level also for honey what is the meaning of …. To contextualize these findings….
Row 549… 0.1379
Better 0.138?
Rows 574-576 Probably a metal analysis on flowers, at least acacia and chestnut, could have explained these values….
Row 586-593 Better a summary table
Figure 3 Poor significance….
Rows 616-621 Better a summary table
Rows 629-631 From what was said in the introduction it is certainly a good idea
Rows 638-651 Better a summary table
Rows 667-668 It would seem obvious…
Roes 693-701 Exactly, the simple evidence of a correlation is not a result or a new knowledge…..
Row 702 Evaluation
Row 727 But there is a cause or reason for the higher BFA of Cr in comparison with other metals?
Rows 738 These are not “conclusions” but results rewritten with the metal amount variations generically attributed to different sources with no significant discussion or data to support them.

Author Response
Reviewer 3
The suggestions of reviewer 3 were made in red to stand out.
Rows 54-66 These rows can be deleted
Row 57 …..ailments
Aliments
Rows 69-75 These rows can be deleted….are too generic
Rows 79-80 These rows can be deleted ….are too generic
Rows 105-121 These rows can be deleted….. are too generic
Rows 149-152 These rows can be deleted….. are too generic
Thank you very much for the suggestion received regarding deleting certain parts in the introduction that are quite obvious and general regarding heavy metals, how they endanger our health, and how they interact with the environment... by removing these general ideas, this manuscript will lose its scientific importance. The realization of this research and the writing of this manuscript was done mainly for the beekeepers in the MaramureÈ™ area, for the honey consumers, but also for the authorities in the field. By deleting these parts of the introduction which for us scientists are very normal...would make it difficult for the reader who is not scientifically qualified to understand the importance of heavy metal pollution. Also, the results of this research will be the basis of new important strategies in this field, so we need all the scientific help obtained from these quite general parts. Once again, thank you very much for your help... regarding the improvement of this manuscript.
Rows 167-170 I would have verified before the chain soil-flower and successively the link soil-flower-honey….The direct link soil-honey with all the problems reported in your introduction, rows 89-97, seems too difficult to study also for the aim of rows 173-175.
Thank you very much for your attention...at the moment we are doing more research to understand much better how the traceability of heavy metals from the soil-flower-honey level works... In the research we are carrying out now...we have collected samples of soil, plants, bees, wax, brood, and honey...so that we can get a clearer picture of the concentration and behavior of heavy metals in the soil system of honey plant.
Row 178 A total of thirty-eight soil samples, designated as S1-S11, and thirty-nine Apis mellifera honey samples, denoted as H1-S11
I thought S1-S38 and ……H1-H39
It has been modified so that the soil samples are marked with S (S1-S11) and the honey samples are marked with H (H1-H11)...so that there is no more confusion regarding the name of the analyzed samples. Also, here it was modified so that the total number of honey samples is 38.
Rows 185-188 All the beehives were away from pollution sources and therefore were all honey’s samples of these beehives?
Yes, the main purpose of this research is to evaluate the degree of pollution of the soil and honey... from different locations located at different distances from the pollution area.
Rows 203-205 Did soil background (and honey?) “produce” only polyfloral honey?
Supplementary Table S1 provides additional details about the geographic origin of the soil samples, sampling depth, anthropogenic impact, and the approximate distance of the hives from pollution sources. In order to understand the phenomenon of pollution both at the level of the soil and in honey, areas were chosen where heavy metal pollution was not reported...in this case it is the background area for the soil. We also have an area that is a background area for honey... and here we have samples of polyfloral honey.
Rows 214-217 Have You evidenced differences between manual or mechanical extraction?
Regarding the method of honey extraction...in this research, honey samples obtained by manual or mechanized extraction were used. The way in which the extraction was carried out is declared by each individual producer. Depending on the amount of honey obtained, an extraction method was chosen...so if the amount of honey is large, a mechanized extraction method was chosen...and if the amount of honey is small, manual extraction was chosen. Regarding the evidence in which the extraction method influences the concentration of heavy metals in honey is obtained based on the DUNCAN statistical analysis presented in tables S8 and S9.
Rows 246-252 Therefore it is always better to treat all honey sample by breaking the organic matrix (acid treatment in microwave or at 550°C).
Yes...in the research already published, the treat all honey sample by breaking the organic matrix (acid treatment in microwave or at 550°C) is recommended so as to obtain the best possible repeatability and reproducibility.
Rows 253-259 It would be better at the start of paragraph, before rows 246-252.
Thank you very much for the suggestion...but by modifying it, we should revise the entire manuscript on the part of bibliographic sources. In future research, we will consider this suggestion.
Rows 259-264 I don’t understand these rows…..in the following (264-267) you treated directly honey in microwave……
The honey samples were placed in Teflon vessels of the digestion system...over which we placed the presented reagents.
Row 269…. Basi ICP-MS Analytical Instrumental Parameters
Basic?
Yes... has been modified.
Row 290 …. 99.99%
Better if you write Ar 5.0 and He 6.0 or 99.999 and 99.9999…………
Yes... has been modified.
Table S5 Three or four decimal figures? LoQ (also LoD) and BEC values are based on different calculations: What have you considered? In the text BEC is not considered……
In Table S5 are presented data regarding LoQ (also LoD) and the values presented are exactly those presented by the ICP-MS software...without any modification...not even rounding or approximations...For analysts and ICP-MS users, the BEC is simply the blank value expressed in concentration units. It is generally obtained by dividing the signal in counts/second (c/s) obtained when aspirating a blank by the slope of the calibration curve in c/s/ppt. Essentially the lower the BEC the easier it is to distinguish an element signal from the background. The important parameter here is the stability of the BEC because the limit of detection is usually calculated as 3 standard deviations of the BEC. Many analysts believe that the BEC provides a more accurate indication of ICP-MS system performance than the limit of detection. A reference to table S5 was added to the manuscript.
Row 339 … test portion of the sample…
Soil? Honey? Where are the results?
These analytical tests were performed directly on soil samples and honey samples...and the results obtained were used to confirm that the digestion and calibration method is the correct one. Unfortunately, these results are the property of the laboratory... and that has more to do with checking the equipment used than the main purpose of this research.
Row 342 …wine sample...
Why not honey?
Manuscript drafting error... this part has been revised.
Figure S1 and S2 They can be deleted…better a table
Thank you very much for the suggestion received...but another reviewer suggests both keeping them and improving them... The role of this image is to highlight the distribution of heavy metals in soil and honey, depending on the sample collection area, I intentionally set the image so that the image is not overloaded...but to distinguish the analyzed samples depending on the metal concentration, we recommend using the tables. In future research, we will use a more efficient and simplified way for the graphic presentation of the results. The legend of the x and y axes have been added to both figures…in the title so that they are as visible as possible without loading the graph even more.
Rows 365-368 Without standard deviation values the two decimal figures don’t seem significant….i.e. Cu 3286.65…..better 3286…..
Changes were made according to the suggestions received
Rows 368-370 …Mercury (Hg) with concentrations below the limit of detection (BLD) which was equal to or greater than the limit of quantification (LoQ) set at 0.1379 μg/L for Hg.
What is the meaning? Limit of detection equal or greater of limit of quantification???? Normally is the reverse…..
The concentration of mercury, in this case, is zero...but according to the quality manual of the laboratory, but also of the laboratory's international accreditation authority, it is not indicated to state that an element has a concentration of 0, but that the respective element has a concentration lower than the limit of quantification.
Rows 370-374 Why not in the table 1?
At the moment, table 1 is quite loaded with a multitude of data and information... the addition of other statistical information would make its reading and understanding extremely difficult... for this reason, we decided to present this information separately from table 1.
Rows 381-398 Better the information of table 1….
Changes were made according to the suggestions received
Row 383 As above (rows 365-368)
Changes were made according to the suggestions received
Rows 405-6
How much?
This information is presented in Table 1 but also in Figure 1...the background areas are Groșii Țibleșului and Viseu de Sus...comparing the results of the analyzes obtained for these areas and their reporting to the results obtained for the areas intensely polluted with heavy metals, a significant difference.
Rows 407-419 It would be sufficient for the references a table…In the soil cultivated with vines high Cu values are typical…and the same for others metals…..
The purpose for which these comparisons were presented is to present as broad a picture as possible of the pollution phenomenon in the MaramureÈ™ area...regardless of the direction of land use. In this area, we are not talking about a polluted area... and this pollution is due to the application of phytosanitary treatments used in the cultivation of vines... this area's pollution is caused by mines and the current tailings settling ponds. The same bibliographic sources as those presented in Table 1 are presented.
Rows 439-444 It would be sufficient for the references or a table
The same bibliographic sources as those presented in Table 1 are presented.
Figure 2 Don’t give significance information’s….better a table.
Another reviewer suggests us to bring new information to the figures in the manuscript...if the editor decides that it is better to give up the figures...they will be deleted.
Rows 468-475 and 489-494 It would be sufficient for the references a table
The same bibliographic sources as those presented in Table 1 are presented.
Rows 516-520 It would be sufficient for the references a table
The same bibliographic sources as those presented in Table 1 are presented.
In all Table 1……for example 156.97 ± 47.60 ……. 157 ± 48….the digital figures when are significance.
Based on the DUNCAN statistical analysis, the presented values are statistically guaranteed... as can be seen between the obtained values there are significant differences.
Row 530…. to be below the detection limit (LoQ for Hg: 0.1379 μg/L)
Therefore ….below the quantification limit….
Changes were made according to the suggestions received
Row 532 Table S8 ….Environment
What is the meaning of environment? How is considered?
Based on the DUNCAN statistical analysis presented in Table S8 and Tables S9.
Row 537 … varying degrees…
What is the meaning in this context? Perhaps it is better deleted…
Phrases have been rewritten so that it is as clear as possible.
Rows 539-542 To contextualize these findings, a comparison was made with the maximum permissible contaminant levels for heavy metals in both food (as specified by Commission Regulation (EC) No 1881/2006) and honey (according to Codex Alimentarius and Council Directive 2001/110/EC).
Changes were made according to the suggestions received
If there is a level also for honey what is the meaning of …. To contextualize these findings….
The values obtained for the honey samples were reported to the maximum levels allowed by international legislation.
Row 549… 0.1379
Better 0.138?
Changes were made according to the suggestions received
Rows 574-576 Probably a metal analysis on flowers, at least acacia and chestnut, could have explained these values….
Thank you very much for your attention...at the moment we are doing more research to understand much better how the traceability of heavy metals from the soil-flower-honey level works... In the research we are carrying out now...we have collected samples of soil, plants, bees, wax, brood, and honey...so that we can get a clearer picture of the concentration and behavior of heavy metals in the soil system of honey plant.
Row 586-593 Better a summary table
These values are already presented in the table...they are used for the national and international reporting of the results obtained in this manuscript.
Figure 3 Poor significance….
Figure 3. Illustrates the dispersion of heavy metals concentration (x axis) (mg/kg) based on the geographical origin (y axis) of the honey samples irrespective of the specific collection year, in Table 3 are Spearman’s correlation matrix.
Rows 616-621 Better a summary table
These values are already presented in the table...they are used for the national and international reporting of the results obtained in this manuscript.
Rows 629-631 From what was said in the introduction it is certainly a good idea
In the research we are carrying out now...we have collected samples of soil, plants, bees, wax, brood, and honey...so that we can get a clearer picture of the concentration and behavior of heavy metals in the soil system of honey plant.
Rows 638-651 Better a summary table
Rows 667-668 It would seem obvious…
Yes...these meanings were calculated using the DUNCAN statistical analysis and are presented in the table S8 and S9.
Roes 693-701 Exactly, the simple evidence of a correlation is not a result or a new knowledge…..
Yes... in the research that we are carrying out now...we will consider the suggestions received...so that this first research achieves the maximum scientific value.
Row 702 Evaluation
Changes were made according to the suggestions received
Row 727 But there is a cause or reason for the higher BFA of Cr in comparison with other metals?
A possible explanation for the high BFA values for Cr...is the use of phytosanitary treatments that are not recommended for the treatment of bee colonies...or Cr has a high degree of traceability from the soil level to the flower nectar. In this case, the research will continue.
Rows 738 These are not “conclusions” but results rewritten with the metal amount variations generically attributed to different sources with no significant discussion or data to support them.
The discussions based on the results are presented in the results and discussions section...there are also certain data regarding the reporting of the results obtained in this research to national and international research. In this section, we chose to present very briefly the main results obtained. At the same time, in this section, important discussions are presented based on the results obtained... the way in which the analyzed elements interact, but also the way in which each element has the ability to bioaccumulate.

Round 2
Reviewer 3 Report
The answers are insufficient to change my opinion of the manuscript (reject).
Author Response
Reviewer 3
The suggestions of reviewer 3 were made in red to stand out.
Rows 54-66 These rows can be deleted
Changes were made according to the suggestions received (this part has been deleted)
Row 57 …..ailments
Aliments
Changes were made according to the suggestions received (this part has been deleted)
Rows 69-75 These rows can be deleted….are too generic
Changes were made according to the suggestions received (this part has been deleted)
Rows 79-80 These rows can be deleted ….are too generic
Changes were made according to the suggestions received (this part has been deleted)
Rows 105-121 These rows can be deleted….. are too generic
Changes were made according to the suggestions received (this part has been deleted)
Rows 149-152 These rows can be deleted….. are too generic
Changes were made according to the suggestions received (this part has been deleted)
Rows 167-170 I would have verified before the chain soil-flower and successively the link soil-flower-honey….The direct link soil-honey with all the problems reported in your introduction, rows 89-97, seems too difficult to study also for the aim of rows 173-175.
In numerous scientific articles already published... the link between the chain has already been demonstrated soil-flower-honey (e.g. Tomczyk et al. 2020 DOI: https://doi.org/10.2478/aucft-2020-0005) (Transfer of Some Toxic Metals from Soil to Honey Depending on Bee Habitat Conditions); Bhalchandra et al. 2022 DOI: https://doi.org/10.22271/j.ento.2022.v10.i1d.8949 (Assessment of essential minerals and toxic trace metals in blended raw honey, soil, leaf and flower samples harvested from different locations of Kannad Taluka of Aurangabad District); Czipa et al. 2017 DOIhttps://doi.org/10.1007/s10661-017-6121-1 (Examination of honeys and flowers as soil element indicators). Thank you very much for your attention...at the moment we are doing more research to understand much better how the traceability of heavy metals from the soil-flower-honey level works... In the research we are carrying out now...we have collected samples of soil, plants, bees, wax, brood, and honey...so that we can get a clearer picture of the concentration and behavior of heavy metals in the soil system of honey plant.
Row 178 A total of thirty-eight soil samples, designated as S1-S11, and thirty-nine Apis mellifera honey samples, denoted as H1-S11
I thought S1-S38 and ……H1-H39
It has been modified so that the soil samples are marked with S (S1-S11) and the honey samples are marked with H (H1-H11)...so that there is no more confusion regarding the name of the analyzed samples. Also, here it was modified so that the total number of honey samples is 38.
Rows 185-188 All the beehives were away from pollution sources and therefore were all honey’s samples of these beehives?
Yes, the main purpose of this research is to evaluate the degree of pollution of the soil and honey... from different locations located at different distances from the pollution area.
Rows 203-205 Did soil background (and honey?) “produce” only polyfloral honey?
Supplementary Table S1 provides additional details about the geographic origin of the soil samples, sampling depth, anthropogenic impact, and the approximate distance of the hives from pollution sources. In order to understand the phenomenon of pollution both at the level of the soil and in honey, areas were chosen where heavy metal pollution was not reported...in this case it is the background area for the soil. We also have an area that is a background area for honey... and here we have samples of polyfloral honey.
Rows 214-217 Have You evidenced differences between manual or mechanical extraction?
Regarding the method of honey extraction...in this research, honey samples obtained by manual or mechanized extraction were used. The way in which the extraction was carried out is declared by each individual producer. Depending on the amount of honey obtained, an extraction method was chosen...so if the amount of honey is large, a mechanized extraction method was chosen...and if the amount of honey is small, manual extraction was chosen. Regarding the evidence in which the extraction method influences the concentration of heavy metals in honey is obtained based on the DUNCAN statistical analysis presented in tables S8 and S9.
Rows 246-252 Therefore it is always better to treat all honey sample by breaking the organic matrix (acid treatment in microwave or at 550°C).
Yes...in the research already published, the treat all honey sample by breaking the organic matrix (acid treatment in microwave or at 550°C) is recommended so as to obtain the best possible repeatability and reproducibility.
Rows 253-259 It would be better at the start of paragraph, before rows 246-252.
Changes were made according to the suggestions received
Rows 259-264 I don’t understand these rows…..in the following (264-267) you treated directly honey in microwave……
The honey samples were placed in Teflon vessels of the digestion system...over which we placed the presented reagents.
Row 269…. Basi ICP-MS Analytical Instrumental Parameters
Basic?
Yes... has been modified.
Row 290 …. 99.99%
Better if you write Ar 5.0 and He 6.0 or 99.999 and 99.9999…………
Yes... has been modified.
Table S5 Three or four decimal figures? LoQ (also LoD) and BEC values are based on different calculations: What have you considered? In the text BEC is not considered……
In Table S5 are presented data regarding LoQ (also LoD) and the values presented are exactly those presented by the ICP-MS software...without any modification...not even rounding or approximations...For analysts and ICP-MS users, the BEC is simply the blank value expressed in concentration units. It is generally obtained by dividing the signal in counts/second (c/s) obtained when aspirating a blank by the slope of the calibration curve in c/s/ppt. Essentially the lower the BEC the easier it is to distinguish an element signal from the background. The important parameter here is the stability of the BEC because the limit of detection is usually calculated as 3 standard deviations of the BEC. Many analysts believe that the BEC provides a more accurate indication of ICP-MS system performance than the limit of detection. A reference to Table S5 was added to the manuscript.
Row 339 … test portion of the sample…
Soil? Honey? Where are the results?
These analytical tests were performed directly on soil samples and honey samples...and the results obtained were used to confirm that the digestion and calibration method is the correct one. Unfortunately, these results are the property of the laboratory... and that has more to do with checking the equipment used than the main purpose of this research.
Row 342 …wine sample...
Why not honey?
Manuscript drafting error... this part has been revised.
Figures S1 and S2 They can be deleted…better a table
Changes were made according to the suggestions received (this part has been deleted)
Rows 365-368 Without standard deviation values the two decimal figures don’t seem significant….i.e. Cu 3286.65…..better 3286…..
Changes were made according to the suggestions received
Rows 368-370 …Mercury (Hg) with concentrations below the limit of detection (BLD) which was equal to or greater than the limit of quantification (LoQ) set at 0.1379 μg/L for Hg.
What is the meaning? Limit of detection equal or greater of limit of quantification???? Normally is the reverse…..
The concentration of mercury, in this case, is zero...but according to the quality manual of the laboratory, but also of the laboratory's international accreditation authority, it is not indicated to state that an element has a concentration of 0, but that the respective element has a concentration lower than the limit of quantification.
Rows 370-374 Why not in the table 1?
Changes were made according to the suggestions received
Rows 381-398 Better the information of table 1….
Changes were made according to the suggestions received
Row 383 As above (rows 365-368)
Changes were made according to the suggestions received
Rows 405-6
How much?
This information is presented in Table 1 but also in Figure 1...the background areas are Groșii Țibleșului and Viseu de Sus...comparing the results of the analyzes obtained for these areas and their reporting to the results obtained for the areas intensely polluted with heavy metals, a significant difference.
Rows 407-419 It would be sufficient for the references a table…In the soil cultivated with vines high Cu values are typical…and the same for others metals…..
The purpose for which these comparisons were presented is to present as broad a picture as possible of the pollution phenomenon in the MaramureÈ™ area...regardless of the direction of land use. In this area, we are not talking about a polluted area... and this pollution is due to the application of phytosanitary treatments used in the cultivation of vines... this area's pollution is caused by mines and the current tailings settling ponds. The same bibliographic sources as those presented in Table 1 are presented.
Rows 439-444 It would be sufficient for the references or a table
The same bibliographic sources as those presented in Table 1 are presented.
Figure 2 Don’t give significance information’s….better a table.
Changes were made according to the suggestions received (this part has been deleted)
Rows 468-475 and 489-494 It would be sufficient for the references a table
The same bibliographic sources as those presented in Table 1 are presented.
Rows 516-520 It would be sufficient for the references a table
The same bibliographic sources as those presented in Table 1 are presented.
In all Table 1……for example 156.97 ± 47.60 ……. 157 ± 48….the digital figures when are significance.
Based on the DUNCAN statistical analysis, the presented values are statistically guaranteed... as can be seen between the obtained values there are significant differences.
Row 530…. to be below the detection limit (LoQ for Hg: 0.1379 μg/L)
Therefore ….below the quantification limit….
Changes were made according to the suggestions received
Row 532 Table S8 ….Environment
What is the meaning of environment? How is considered?
Based on the DUNCAN statistical analysis presented in Table S8 and Tables S9.
Row 537 … varying degrees…
What is the meaning in this context? Perhaps it is better deleted…
Phrases have been rewritten so that it is as clear as possible.
Rows 539-542 To contextualize these findings, a comparison was made with the maximum permissible contaminant levels for heavy metals in both food (as specified by Commission Regulation (EC) No 1881/2006) and honey (according to Codex Alimentarius and Council Directive 2001/110/EC).
Changes were made according to the suggestions received
If there is a level also for honey what is the meaning of …. To contextualize these findings….
The values obtained for the honey samples were reported to the maximum levels allowed by international legislation.
Row 549… 0.1379
Better 0.138?
Changes were made according to the suggestions received
Rows 574-576 Probably a metal analysis on flowers, at least acacia and chestnut, could have explained these values….
Thank you very much for your attention...at the moment we are doing more research to understand much better how the traceability of heavy metals from the soil-flower-honey level works... In the research we are carrying out now...we have collected samples of soil, plants, bees, wax, brood, and honey...so that we can get a clearer picture of the concentration and behavior of heavy metals in the soil system of honey plant.
Row 586-593 Better a summary table
These values are already presented in the table...they are used for the national and international reporting of the results obtained in this manuscript.
Figure 3 Poor significance….
Figure 3. Illustrates the dispersion of heavy metals concentration (x axis) (mg/kg) based on the geographical origin (y axis) of the honey samples irrespective of the specific collection year, in Table 3 are Spearman’s correlation matrix.
Rows 616-621 Better a summary table
These values are already presented in the table...they are used for the national and international reporting of the results obtained in this manuscript.
Rows 629-631 From what was said in the introduction it is certainly a good idea
In the research we are carrying out now...we have collected samples of soil, plants, bees, wax, brood, and honey...so that we can get a clearer picture of the concentration and behavior of heavy metals in the soil system of honey plant.
Rows 638-651 Better a summary table
Changes were made according to the suggestions received
Rows 667-668 It would seem obvious…
Yes...these meanings were calculated using the DUNCAN statistical analysis and are presented in the table S8 and S9.
Roes 693-701 Exactly, the simple evidence of a correlation is not a result or a new knowledge…..
Yes... in the research that we are carrying out now...we will consider the suggestions received...so that this first research achieves the maximum scientific value.
Row 702 Evaluation
Changes were made according to the suggestions received
Row 727 But there is a cause or reason for the higher BFA of Cr in comparison with other metals?
A possible explanation for the high BFA values for Cr...is the use of phytosanitary treatments that are not recommended for the treatment of bee colonies...or Cr has a high degree of traceability from the soil level to the flower nectar.
Rows 738 These are not “conclusions” but results rewritten with the metal amount variations generically attributed to different sources with no significant discussion or data to support them.
The conclusions part has been rewritten
The study analyzed 38 soil samples from the MaramureÈ™ region, focusing on nine potentially toxic elements at a 0-10 cm depth. The highest concentrations were Copper (Cu) at 3286.65 mg/kg, Zinc (Zn) at 2834.58 mg/kg, Lead (Pb) at 1205.57 mg/kg, Chromium (Cr) at 10.75 mg/kg, Nickel (Ni) at 7.99 mg/kg, Cadmium (Cd) at 6.33 mg/kg, Cobalt (Co) at 5.98 mg/kg, Arsenic (As) at 4.37 mg/kg, and Mercury (Hg) below the detection limit. Soil samples near anthropogenic areas, like mining operations and settling ponds, had significantly higher metal concentrations, with Aurul settling pond and Herja mine being major sources. Copper and Zinc exceeded legal limits in some areas, posing environmental risks. Acacia honey had the highest Copper levels, influenced by nearby mining sites. Lead and Cadmium concentrations exceeded legal limits in certain areas, likely due to the Aurul settling pond's influence. The study emphasizes stricter adherence to environmental regulations.
The research also examined heavy metal concentrations in different honey types (chestnut, acacia, polyfloral) from various locations. Chromium had the highest mean concentration at 0.58 mg/kg, followed by Zinc (0.56 mg/kg), Copper (0.34 mg/kg), Lead (0.10 mg/kg), Nickel (0.03 mg/kg), Cadmium (0.01 mg/kg), and Cobalt below the detection limit. Arsenic and Mercury were undetected. Copper concentrations in honey were notably high in the Baia Mare area, mainly due to former mining sites and settling ponds. Zinc levels exceeded international limits in some areas, warranting hive relocation. Acacia honey had the highest Lead and Cadmium concentrations, surpassing permissible levels. The study highlights the importance of monitoring heavy metal contamination in honey for human and environmental safety.
The analysis found significant correlations between toxic elements in honey, with notable positive correlations for Zn/Cu, Pb/Cu, Pb/Zn, Cd/Cu, Cd/Zn, Ni/Cu, and Ni/Cd. A negative correlation was identified between Cr/Cu. Hive location and area, year/type of honey, and proximity to pollution sources also influenced metal concentrations in honey. Cu/area had a strong negative correlation. Pb/honey type and Ni/honey type showed negative correlations. Cr concentrations were influenced by these factors and Zn interaction. Further investigations are needed to validate these relationships, considering local agricultural practices, geological variations, and alternative pollution sources. The study employed the Bioaccumulation Factor (BAF) to assess metal transfer from soil to honey. The sequential transfer of metals from soil to honey was Cr > Ni > Cd > Zn > Pb > Cu. BAF analysis by honey type and production area revealed variation in metal accumulation patterns, with Cr accumulation raising concerns. This region showed signs of heavy metal pollution, particularly Cd, Cu, Pb, and Zn. Unexpectedly, Cr pollution was observed, possibly due to nearby agricultural phytosanitary treatments.
